# Analytical Construction on Geometric Architectures: Transitioning from Static to Temporal Link Prediction

Yadong Sun [1]  Xiaofeng Cao [1 2]  Ivor W. Tsang [3 4]  Heng Tao Shen [2 5]

## Abstract

Static systems exhibit diverse structural properties, such as hierarchical, scale-free, and isotropic patterns, where different geometric spaces offer unique advantages. Methods combining multiple geometries have proven effective in capturing these characteristics. However, real-world systems often evolve dynamically, introducing significant challenges in modeling their temporal changes. To overcome this limitation, we propose a unified cross-geometric learning framework for dynamic systems, which synergistically integrates Euclidean and hyperbolic spaces, aligning embedding spaces with structural properties through fine-grained substructure modeling. Our framework further incorporates a temporal state aggregation mechanism and an evolution-driven optimization objective, enabling comprehensive and adaptive modeling of both nodal and relational dynamics over time. Extensive experiments on diverse real-world dynamic graph datasets highlight the superiority of our approach in capturing complex structural evolution, surpassing existing methods across multiple metrics.

## 1. Introduction

In recent years, static graph neural networks (GNNs) in different geometric spaces have achieved remarkable progress in the field of link prediction (Yang et al., 2022; Chen et al., 2023). For graphs with general structural features, GNNs in Euclidean space demonstrate superior performance (Wu et al., 2022), while for graphs with specialized structures,

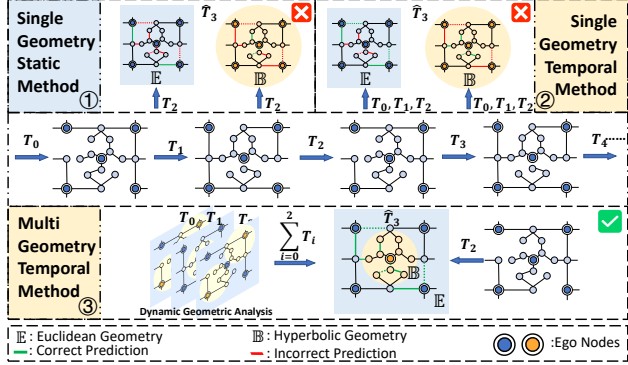

Figure 1: Dynamic graph requires continuous and independent modeling of snapshots at each timestamp, and the geometric properties of local structures may vary with link evolution. Existing methods typically apply single geometry analysis at static timestamps (①) or across time sequences (②), leading to mismatches between geometric heterogeneity and data, causing modeling distortions. Therefore, A temporal continuous multi geometry analysis method (③) that selects different geometric embeddings for distinct local structures in each snapshot is more universal and effective.

such as hierarchical structures, GNNs in hyperbolic space emerge as the preferred choice (Peng et al., 2021). Nevertheless, static GNNs are inherently limited in capturing the evolving nature of relationships in real-world systems, where interactions between entities—such as in social interactions (Liao et al., 2018; Huang et al., 2021), scientific collaborations (Yasunaga et al., 2019), and transportation networks (Yu et al., 2017; Ji et al., 2023)—are dynamic and continuously change over time, posing significant challenges for static models (Han et al., 2021). Temporal link prediction aims to forecast the appearance of new links and the potential disappearance of existing links based on current and past graph states (Divakaran & Mohan, 2020). With the expansion of network scale and increasing complexity of link evolution, accurate temporal link prediction is crucial not only for enhancing the understanding of dynamic networks but also for improving the performance and decision quality of intelligent systems (Zhang et al., 2023).

Conventional GNN methods are designed for static graph

[1]School of Artificial Intelligence, Jilin University, China. [2]School of Computer Science and Technology, Tongji University, China. [3]CFAR and IHPC, Agency for Science, Technology and Research (A*STAR), Singapore. [4]College of Computing and Data Science, NTU, Singapore [5]School of Computer Science and Engineering, University of Electronic Science and Technology of China. Correspondence to: Xiaofeng Cao <xiaofengcao@jlu.edu.cn>.

*Proceedings of the 42nd International Conference on Machine Learning*, Vancouver, Canada. PMLR 267, 2025. Copyright 2025 by the author(s).

structures and cannot adapt to the evolving nature of dynamic graphs (Zhu et al., 2020; Xu et al., 2022a; Sun et al., 2024). When links appear or disappear, static models struggle to dynamically adjust embeddings, resulting in limited suitability for dynamic graph tasks. While dynamic GNNs have been proposed to model temporal variations using a single geometric space (Hajiramezanali et al., 2019; Pareja et al., 2020; Bai et al., 2023), they neglect the geometric heterogeneity across varying local structures (Gu et al., 2018; Bachmann et al., 2020), limiting their ability to capture diverse characteristics in dynamic graphs (Ravasz & Barabási, 2003; Bronstein et al., 2017). Furthermore, the evolution of links over time alters local geometric properties, a critical aspect overlooked by existing methods. Thus, there is a need for approaches that integrate geometric heterogeneity with temporal dynamics, enabling models to dynamically select appropriate geometric embedding and adjust them in response to link evolution (Zhou et al., 2023), thereby capturing the temporal evolution of graph data more effectively.

**Limitations**: Drawing upon the preceding analysis, we have identified two fundamental limitations in the existing link prediction methods: 1) Limited Flexibility in Geometric Representation: Existing approaches often rely on fixed geometric spaces, which may not fully accommodate the complex, hierarchical, or non-Euclidean local structures present in diverse temporal graph data. 2) Inadequate Consideration of Temporal Dynamics: Many current methods fail to effectively capture the intricate temporal dependencies and patterns in evolving links of dynamic graphs, leading to suboptimal predictive performance over time. Below, we present our observations regarding these limitations: 1) Current dynamic models predominantly embed graphs into a single geometric space, which overlooks the geometric diversity inherent in local structures. Hierarchical subgraphs are better suited to hyperbolic space, while non-hierarchical ones align more effectively with Euclidean space (Zhu et al., 2020; Gu et al., 2019; Shang et al., 2024). 2) These models mainly focus on macro-level changes, often overlooking micro-level link variations across timestamps, which leads to a lack of mechanisms to fully capture these nuances.

**Schemes**: To relieve the limitations mentioned above, we propose a framework conducts dynamic analysis of temporal link prediction by integrating multiple geometric spaces. Specifically, to integrate dynamic temporal information into geometric analysis, it continuously extracts $k$-hop egographs centered on key nodes at each timestamp, adapting the geometric representation based on the evolving local structural patterns over time. It then aligns and optimizes hierarchical embeddings from both non-Euclidean and Euclidean spaces, using multi geometric information as graph features for the current timestamp. Additionally, the framework maps the representations from previous timestamps to a high-dimensional space, computes attention coefficients

for each past timestamp, and aggregates them to obtain the input hidden states for the current timestamp. Finally, a link evolution loss function is used to capture fine-grained link dynamics by optimizing distances between node pairs corresponding to newly appeared and disappeared links.

The salient aspects of our contributions are as follows:

- Geometric analysis over time series characterizes intricate structural evolution, capturing fine-grained topological variations and revealing the temporal transformation of geometric features.

- Building upon this more refined consciousness, a fine-grained modeling framework for dynamic graph features is proposed, capturing complex structural variations through state aggregation across different geometric domains. Additionally, a link evolution loss function is designed to precisely characterize link changes.

- Demonstrated the effectiveness of the dynamic geometric analysis paradigm through comprehensive evaluation across datasets of varying scales and types.

## 2. Related Work

**Geometric Graph Learning** Graph-based models have gained prominence due to their ability to represent relational data effectively (Wu et al., 2020; Cheng et al., 2023). After the introduction of GCN (Kipf & Welling, 2017), many message-passing based GNNs (Gao & Ji, 2019; Song et al., 2020; Wan et al., 2021; Huo et al., 2023) were proposed, achieving excellent performance across various graph learning tasks. GAT (Veličković et al., 2018) enhances node representation by using attention mechanisms to weigh neighboring nodes differently. GraphSAGE (Hamilton et al., 2017) efficiently handles large-scale graphs by sampling and aggregating neighboring node features. Graph Transformer (Yun et al., 2019) use self-attention to capture complex dependencies in graphs. These models are all designed in Euclidean space. To address issues related to hierarchical and complex graph structures, some models, such as HGCN (Chami et al., 2019), H2H-GCN (Dai et al., 2021), HAT (Zhang et al., 2021), HypFormer (Yang et al., 2024), are designed in hyperbolic space, allowing for more effective representation of hierarchical and topological relationships within graphs. In recent years, combining the advantages of both geometries has become a new graph learning paradigm (Zhu et al., 2020; Xu et al., 2022b; Shang et al., 2024).

**Temporal Link Prediction** Temporal Link Prediction focuses on forecasting the appearance of future links or the disappearance of existing links in dynamic graphs over time. To efficiently model temporal links, many dynamic graph neural networks with sequential architectures have been proposed (Skarding et al., 2021). GCRN (Seo et al., 2018) combines graph convolution and recurrent layers (Shi

et al., 2015) for enhanced sequence prediction. DySAT (Sankar et al., 2020) captures node representations in evolving graphs using dynamic self-attention. VGRNN (Hajiramezanali et al., 2019) introduces hierarchical variational modeling to capture topology and attribute changes in dynamic graphs. EvolveGCN (Pareja et al., 2020) improves dynamic graph learning by evolving GCN parameters with an RNN. Some methods model dynamic graphs in hyperbolic space, such as HTGN (Yang et al., 2021) and HG-WaveNet (Bai et al., 2023), achieving promising results in capturing hierarchical and complex structures. While as dynamic graphs grow in complexity, the structural heterogeneity of different local structures poses challenges for single-geometric methods. Currently, there are no approaches that utilize multiple geometries for this task.

## 3. Preliminaries

### 3.1. Problem Formulation

Given a dynamic graph $\mathcal{G}_t = (\mathcal{V}, \mathcal{E}_t), t \in \mathbb{T}$, where $\mathbb{T} = \{t_1, t_2, \ldots, t_T\}$ is a discrete timestamp set, $\mathcal{V}$ denotes the set of nodes and $\mathcal{E}_t \subseteq \mathcal{V} \times \mathcal{V}$ denotes the set of edges present at timestamp $t$. For temporal link prediction task, we aim to predict the states of links at a future timestamp $t^+ \in \mathbb{T}$. Predicting the appearance of an edge by estimating the probability that an edge $(u, v)$ exists at timestamp $t^+$:

$$\hat{y}_{uv}(t^+) = \Pr[(u, v) \in \mathcal{E}_{t^+} \mid \mathcal{G}_{\leq t}], \quad (1)$$

where $\mathcal{G}_{\leq t}$ denotes the sequence $\{\mathcal{G}_{t_1}, \mathcal{G}_{t_2}, \ldots, \mathcal{G}_t\}$. Predicting edge disappearance by estimating the probability that an edge $(u, v)$ does not exist at time $t^+$:

$$\hat{y}_{uv}^{\text{absent}}(t^+) = \Pr[(u, v) \notin \mathcal{E}_{t^+} \mid \mathcal{G}_{\leq t}], \quad (2)$$

The overall objective is to develop **a temporal link prediction model $f$** that provides a probability distribution over the existence and non-existence of all potential edges $(u, v)$:

$$f(\mathcal{G}_{\leq t}) \to \{\hat{y}_{uv}(t^+), \hat{y}_{uv}^{\text{absent}}(t^+)\}_{(u,v) \in \mathcal{V} \times \mathcal{V}}, \quad (3)$$

### 3.2. Hyperbolic Geometry

Hyperbolic geometry describes spaces characterized by constant negative curvature, where volume grows exponentially with the increase in spatial dimensions. A $n$-dimensional hyperbolic space is a complete Riemannian manifold with a constant negative curvature $c$, denoted as $(\mathbb{H}_c^n, g^{\mathbb{H}})$, where $g^{\mathbb{H}}$ is the Riemannian metric. Hyperbolic space can be modeled using five isometric models (Beltrami, 1868; Cannon et al., 1997), in this paper, we adopt Poincaré disk model.

**Definition 1 (Poincaré disk model)** *The Poincaré disk model $\mathbb{B}$ is a manifold equipped with a Riemannian metric*

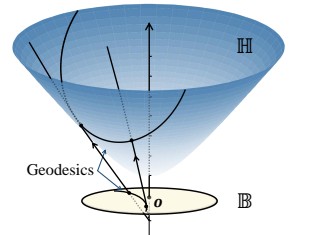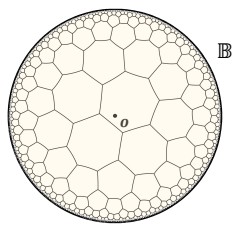

Figure 2: The poincaré disk model $\mathbb{B}$ is given by projecting each point of hyperboloid model $\mathbb{H}$ onto the hyperplane $\boldsymbol{o}$.

$g^{\mathbb{B}}$*, which defined as:*

$$\mathbb{B}_c^n := \{\boldsymbol{x} \in \mathbb{R}^n : -c\|\boldsymbol{x}\|^2 < 1\},$$
$$g^{\mathbb{B}} = \lambda_{\boldsymbol{x}}^2 g^{\mathbb{E}}, \quad \lambda_{\boldsymbol{x}} = \frac{2}{1 - \|\boldsymbol{x}\|^2}, \quad (4)$$

*where $\|\cdot\|$ denotes the Euclidean norm, $g^{\mathbb{E}}$ denotes the Euclidean metric, and the superscript $\mathbb{B}$ indicates that the vector or matrix is in the hyperbolic space modeled using the Poincaré disk model.*

**Definition 2 (Hyperbolic Operations)** *Given two points $\boldsymbol{x}, \boldsymbol{y} \in \mathbb{B}_c^n$, the hyperbolic distance between them is defined by*

$$d_c(\boldsymbol{x}, \boldsymbol{y}) = \frac{2}{\sqrt{c}} \tanh^{-1}\left(\sqrt{c}\,\|-\boldsymbol{x} \oplus_c \boldsymbol{y}\|\right), \quad (5)$$

*where $\oplus_c$ denotes Möbius addition, given by*

$$\boldsymbol{x} \oplus_c \boldsymbol{y} := \frac{\left(1 + 2c\langle \boldsymbol{x}, \boldsymbol{y}\rangle + c\|\boldsymbol{y}\|^2\right)\boldsymbol{x} + \left(1 - c\|\boldsymbol{x}\|^2\right)\boldsymbol{y}}{1 + 2c\langle \boldsymbol{x}, \boldsymbol{y}\rangle + c^2\|\boldsymbol{x}\|^2\|\boldsymbol{y}\|^2}, \quad (6)$$

$\langle \boldsymbol{x}, \boldsymbol{y}\rangle$ *denotes the Euclidean inner product of $\boldsymbol{x}$ and $\boldsymbol{y}$.*

**Definition 3 (Tangent Space)** *The tangent space at a point $\boldsymbol{x}$ in hyperbolic space, denoted $\mathcal{T}_{\boldsymbol{x}}\mathbb{B}_c^n$, approximates the local structure of the space to first order. This $n$-dimensional tangent space is isometric to Euclidean space $\mathbb{R}^n$. The mapping between hyperbolic space and the tangent space is facilitated by the exponential and logarithmic maps, defined as follows:*

$$\exp_{\boldsymbol{x}}^c(\boldsymbol{v}) = \boldsymbol{x} \oplus_c \left(\tanh\left(\sqrt{c}\frac{\lambda_{\boldsymbol{x}}^c\|\boldsymbol{v}\|}{2}\right)\frac{\boldsymbol{v}}{\sqrt{c}\|\boldsymbol{v}\|}\right), \quad (7)$$

$$\log_{\boldsymbol{x}}^c(\boldsymbol{y}) = d_c(\boldsymbol{x}, \boldsymbol{y})\frac{-\boldsymbol{x} \oplus_c \boldsymbol{y}}{\lambda_{\boldsymbol{x}}^c\|-\boldsymbol{x} \oplus_c \boldsymbol{y}\|}, \quad (8)$$

*where $\boldsymbol{v} \in \mathcal{T}_{\boldsymbol{x}}\mathbb{B}_c^n$, $\boldsymbol{y} \in \mathbb{B}_c^n$ and $\lambda_{\boldsymbol{x}}^c$ has same meaning in Eq. (4). To ensure consistency in error metrics across various directions, we use the origin $\boldsymbol{o}$ in hyperbolic space as the reference point $\boldsymbol{x}$.*

## 4. Methodology

In this section, we introduce the key components of the framework and its pipeline (Figure 3). We also detail the

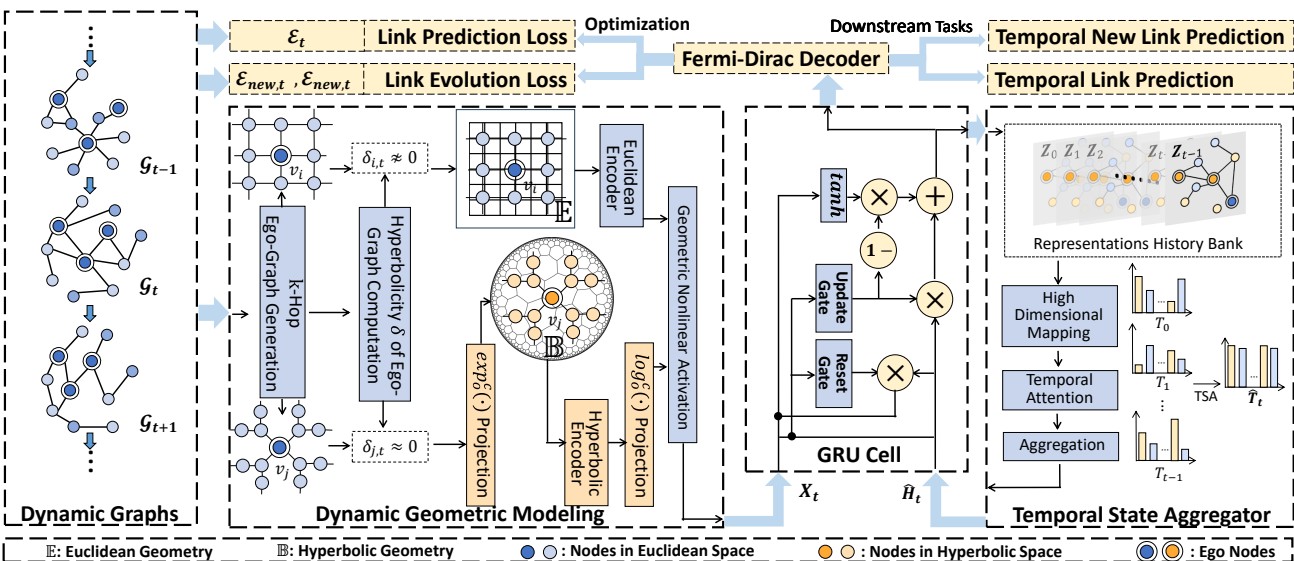

Figure 3: Architecture of our method. Given the graph $G_t$ at timestamp $t$, the Dynamic Geometric Modeling (DGM) module calculates each local structure's hyperbolicity $\delta$ via a k-hop ego-graph, selecting and optimizing the appropriate embedding space. The Temporal State Aggregator (TSA) layer then predicts the current hidden state by aggregating past representations using high-dimensional mapping and attention coefficients. Finally, a GRU cell processes the current representation, and link prediction is performed using the Fermi-Dirac decoder, with the model optimized via Link Evolution Loss (LEL) and traditional link prediction loss functions.

optimization process and complexity analysis of the full framework.

### 4.1. Dynamic Geometric Modeling

To capture the geometric heterogeneity inherent in different local structures of dynamic graphs, Dynamic Geometric Modeling (DGM) module is designed to model various local structures of the graph at each timestamp into geometric spaces that best align with their inherent characteristics, thereby uncovering the latent geometric relationships and hierarchical features. This module enhances the representation of complex structures and interactions, improving the ability of overall model to accurately predict and analyze dynamic link evolution over time.

**Definition 4 (k-Hop Ego-Graph of Centroid $v$)** *For a given node $v \in \mathcal{V}$, its corresponding $k$-hops ego-subgraph $\mathcal{G}_v$ comprises all nodes $w \in \mathcal{V} \backslash \{v\}$ within a distance no greater than $k$ from $v$, along with their respective links.*

Let $\mathcal{G}_t = (\mathcal{V}_t, \mathcal{E}_t)$ denote snapshot of the graph at timestamp $t$. According to *the local subgraph preservation property* (Huang & Zitnik, 2020), the influence of neighbors on centroid decays exponentially with radius increasing. The local structural characteristics of a node can be captured by its $k$-hop ego-graph. Thus, we samples $k$-hop ego-graphs, and computes their hyperbolicities $\delta_{i,t}$ [1] at $t$.

---

[1] Gromov's $\delta$-hyperbolicity (Adcock et al., 2013; Gromov,

For node $v_{i,t}$, its hyperbolicity $\delta_{i,t}$ can be computed by:

$$\delta_{i,t} = \max_{u_1, u_2, u_3, u_4 \in \mathcal{V}_{\mathcal{G}_{i,t}}} \frac{\delta(u_1, u_2, u_3, u_4)}{D}, \quad (9)$$

where $u_1, u_2, u_3, u_4$ denote node quadruplets from ego-subgraph $\mathcal{G}_{i,t}$, $D$ denotes the maximum shortest path length between any two nodes among them. $\delta(u_1, u_2, u_3, u_4) = \ell(u_1, u_2) + \ell(u_3, u_4) - \ell(u_1, u_3) + \ell(u_2, u_4)$, if shortest path lengths between node pairs satisfy $\ell(u_1, u_2) + \ell(u_3, u_4) \geq \ell(u_1, u_3) + \ell(u_2, u_4) \geq \ell(u_1, u_4) + \ell(u_2, u_3)$.

As illustrated in (Adcock et al., 2013), structural features of ego-graphs become significantly apparent when the diameter of the quadruplets exceeds 2. When the diameter increases to 5, the resulting features become similar. Hence, to obtain the most accurate hyperbolicity values for local structures, we set the $k$ value to 4.

For tree-like hierarchical local structures ($\delta_{i,t} \approx 0$), we first map them into the hyperbolic space. Assuming that the features $\mathbf{F}^{\mathbb{E}}_{\delta_{i,t} \approx 0}$ of the local structure lie in the tangent space at the origin $o$ of the Poincaré disk model, where the superscript $\mathbb{E}$ denotes that the features are in Euclidean space. The features $\mathbf{F}^{o,\mathbb{E}}_{\delta_{i,t} \approx 0}$ are then mapped into the hyperbolic space via exponential map:

$$\mathbf{X}^{\mathbb{B}}_{\delta_{i,t} \approx 0} = \exp^c_o(\mathbf{X}^{o,\mathbb{E}}_{\delta_{i,t} \approx 0}). \quad (10)$$

---

1987) measures a graph's tree-like structure; lower $\delta$ values indicate higher hyperbolicity, $\delta = 0$ representing a tree.

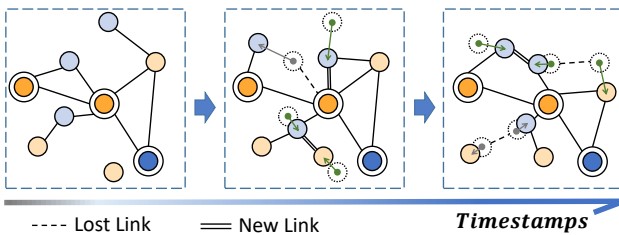

---- Lost Link    ═══ New Link    *Timestamps*

Figure 4: Link Evolution Loss adjusts the distribution of nodes based on edges added and removed between the current and previous snapshots, dynamically capturing changes in relationships between nodes.

Then, we process the features using HGCN (Chami et al., 2019) to obtain the corresponding hyperbolic embeddings:

$$\mathbf{F}^{\mathbb{B}}_{\delta_{i,t}\approx 0} = \exp_o^c(\mathbf{W}^{\mathbb{B}} \log_o^c(\mathbf{X}^{\mathbb{B}}_{\delta_{i,t}\approx 0})) \oplus_c \boldsymbol{b}^{\mathbb{B}}, \qquad (11)$$

where $\mathbf{W}^{\mathbb{B}}$ denotes weight matrix, $\boldsymbol{b}^{\mathbb{B}}$ denotes bias vector.

For the remain local structures ($\delta_{i,t} \not\approx 0$), we use GCN (Kipf & Welling, 2017) to obtain corresponding Euclidean embeddings:

$$\mathbf{F}^{\mathbb{E}}_{\delta_{i,t}\not\approx 0} = \mathbf{W}^{\mathbb{E}} \cdot \mathbf{X}^{\mathbb{E}}_{\delta_{i,t}\not\approx 0} + \boldsymbol{b}^{\mathbb{E}}. \qquad (12)$$

We approximate the optimal embedding space distribution nonlinear optimization function using a multi-layer perceptron, and ultimately obtain the final cross-geometric embedding for the current timestamp as:

$$\mathbf{X_t} = \sigma^{\otimes^c}(\mathbf{W}^f \cdot (\log_o^c(\mathbf{F}^{\mathbb{B}}_{\delta_{i,t}\approx 0}) \,||\, \mathbf{F}^{\mathbb{E}}_{\delta_{i,t}\not\approx 0} + \boldsymbol{b}^f), \quad (13)$$

where $\sigma^{\otimes^c}$ denotes geometric non-linear activation with different curvatures.

### 4.2. Temporal State Aggregator

Temporal State Aggregator (TSA) is a more straightforward and efficient network layer compared to existing methods, designed to forecast the hidden state at the subsequent timestamp by synthesizing temporal information from historical states. For a sequence of representations $\{\mathbf{Z}_0, \mathbf{Z}_1, \ldots, \mathbf{Z}_{t-1}\}$, where each $\mathbf{Z}_t$ denotes the representations of dynamic graph $\mathcal{G}_t$ at timestamp $t$. TSA initially transforms these representations into a higher-dimensional space using a nonlinear mapping function to enhance the distinguishability between states at different timestamps:

$$\mathbf{Z}_{tsa} = \phi(\mathbf{W}_{\text{tsa}} \cdot ||_{i=0}^{t-1}\mathbf{Z}_i + \mathbf{b}_{\text{tsa}}), \qquad (14)$$

where $\phi$ is a nonlinear activation function such as the hyperbolic tangent function, and $\mathbf{W}_{\text{tsa}}$ and $\mathbf{b}_{\text{tsa}}$ are the transformation matrix and bias vector, respectively.

The next step involves calculating temporal attention scores, which assess the importance of each historical state. This is achieved through:

$$\alpha = \text{softmax}\left(\gamma(\mathbf{W}_{\text{att}} \cdot \mathbf{Z}_{tsa} + \mathbf{b}_{\text{att}})\right), \qquad (15)$$

where $\gamma$ denotes a scalar function applied to adjust the scale of the attention scores.

The predicted hidden state at the next timestamps $t$ is then derived by aggregating the historical representations weighted by their respective attention scores:

$$\hat{\mathbf{H}}_t = \sum_{i=0}^{t-1} \alpha_i \cdot \mathbf{Z}_i. \qquad (16)$$

The TSA effectively harnesses the temporal dependencies within the data by emphasizing the most influential historical states. This integration of significant past information facilitates enhanced prediction accuracy for future states while optimizing computational efficiency through advanced tensor operations.

### 4.3. Link Evolution Loss

To model and analyze fine-grained link dynamics, we propose the Link Evolution Loss (LEL) function, which to optimize the spatial arrangement of node embeddings by considering micro-level changes of edges. Specifically, LEL obtain two edge sets, $\mathcal{E}_{\text{new}}$ and $\mathcal{E}_{\text{lost}}$, where $\mathcal{E}_{\text{new}} \subseteq \mathcal{E}_t$ and $\mathcal{E}_{\text{new}} \cap \mathcal{E}_{t-1} = \emptyset$, $\mathcal{E}_{\text{lost}} \subseteq \mathcal{E}_{t-1}$ and $\mathcal{E}_{\text{lost}} \cap \mathcal{E}_t = \emptyset$. Let $\mathbf{F}_t$ and $\mathbf{F}_{t-1}$ denote the cross-space embedding matrices at time $t$ and $t-1$, respectively.

For nodes pairs $(\boldsymbol{u}, \boldsymbol{v})$ of new edges in $\mathcal{G}_t$, the goal of LEL is to bring the distributions of them closer if they are newly be connected. This is achieved by minimizing:

$$\mathcal{L}_{new,t} = \sum_{(u,v)\in\mathcal{E}_{new}} \max\left(0, |sim(f_{u,t}, f_{v,t})| - \tau\right), \;(17)$$

where $sim(\cdot)$ denotes the cosine similarity function, $f_{u,t}$ and $f_{v,t}$ denote embeddings of $\boldsymbol{u}$ and $\boldsymbol{v}$ at timestamp $t$, $\tau$ denotes penalty term, which is typically set to 0.5.

Conversely, for nodes pairs of lost edges, the objective is to increase the distance between distributions of nodes pairs that should no longer be connected:

$$\mathcal{L}_{lost,t} = \sum_{(u,v)\in\mathcal{E}_{lost}} \max\left(0, \tau - |sim(f_{u,t}, f_{v,t})|\right). \;(18)$$

The overall loss function combines these components and applies a sigmoid function for normalization:

$$\mathcal{L}_{LEL,t} = \sigma\left(\mathcal{L}_{\text{new}} + \mathcal{L}_{\text{lost}}\right), \qquad (19)$$

where $\sigma$ denotes the sigmoid function.

## 4.4. Overall Framework

**Workflow:** With aforementioned key components, we can construct the complete framework, as illustrated in Figure 3, the complete algorithmic procedure can be found in Appendix C.1. At timestamp $t$, it inputs the node features of the graph snapshot $\mathcal{G}_t$ into the DGM module, yielding the optimized cross-space feature embedding $\mathbf{X}_t$. Simultaneously, the TSA layer computes the current hidden state input $\hat{\mathbf{H}}_t$ based on the representations from previous timestamps $\{\mathbf{Z}_0, \ldots, \mathbf{Z}_{t-1}\}$. It then feeds $\mathbf{X}_t$ and $\hat{\mathbf{H}}_t$ into the GRU cell to obtain the node representation $\mathbf{Z}_t$ at time $t$. Building upon this, it utilizes a Fermi-Dirac decoder for link prediction, expressed by the following formula:

$$p_{F-D}(\boldsymbol{z}_i, \boldsymbol{z}_j) = \frac{1}{exp(d(\boldsymbol{z}_i, \boldsymbol{z}_j) - r)/s}, \qquad (20)$$

where r and s are hyper-parameters, $\boldsymbol{z}_i$ and $\boldsymbol{z}_j$ denote the representations of the two nodes for which the edge is predicted.

**Optimization:** Our optimization objective is to maximize the prediction accuracy of future unobserved edges, as formalized in Eq. 3. At timestamp $t$, the loss function for link prediction is given by:

$$\mathcal{L}_{LP,t} = \frac{1}{|\mathcal{E}_t|} \Big( \sum_{e_{i,j} \in \mathcal{E}_t} - \log \left( p_{F-D} \left( \boldsymbol{z}_{t,i}, \boldsymbol{z}_{t,j} \right) \right) \\ - \sum_{e_{i'j'} \notin \mathcal{E}_t} \left( 1 - \log \left( p_{F-D} \left( \boldsymbol{z}_{t,i'}, \boldsymbol{z}_{t,j'} \right) \right) \right). \qquad (21)$$

In addition, the framework computes the link evolution loss $\mathcal{L}_{LEL,t}$, which quantifies the microscopic changes in link structure between $G_t$ and $G_{t-1}$. Consequently, the overall loss function at timestamp $t$ is:

$$\mathcal{L} = \mathcal{L}_{LP,t} + \mathcal{L}_{LEL,t}. \qquad (22)$$

By minimizing this combined loss function, the framework simultaneously optimizes both the accuracy of edge predictions and the ability to capture evolving link patterns, thereby enhancing its overall performance in dynamic link prediction tasks.

## 4.5. Complexity Analysis

The computational complexity of our method can be summarized by three modules as follows:

**DGM**: It calculating the hyperbolicity $\delta_{i,t}$ of $k$-hop egographs. The complexity of computing $\delta_{i,t}$ for each node involves evaluating node quadruplets, which is $O(k^4 \cdot N)$, where $N$ is the number of nodes. Additionally, mapping features to hyperbolic space and performing Euclidean or hyperbolic embeddings requires $O(N \cdot (d_{\mathbb{E}} + d_{\mathbb{H}}))$, where

$d_{\mathbb{E}}$ and $d_{\mathbb{H}}$ denote the dimensions of the Euclidean and hyperbolic embeddings, respectively.

**TSA**: Transforming and aggregating previous states states involves a complexity of $O(T \cdot d_t)$, where $T$ is the number of time steps and $d_t$ is the dimensionality of the transformed state representations.

**LEL**: It adjusts node embeddings based on newly added and lost edges. The complexity of computing the link evolution loss is $O(|\mathcal{E}_{new}| + |\mathcal{E}_{lost}|)$, where $|\mathcal{E}_{new}|$ and $|\mathcal{E}_{lost}|$ represent the number of new and lost edges, respectively.

The overall complexity depends on the graph size $N$, number of timestamps $T$, embedding dimensions, and edge evolution. The computational complexity is:

$$O(k^4 \cdot N + N \cdot (d_E + d_H) + T \cdot d_t + |\mathcal{E}_{new}| + |\mathcal{E}_{lost}|). \quad (23)$$

## 5. Experiments

In this section, we conducted comparative experiments, efficiency analysis, hyperparameter analysis, and ablation studies to validate the effectiveness and advantages of our method. Due to space limitations, only key experimental details and results are presented here, with additional results available in Appendix D.

### 5.1. Experimental Setup

**Datasets.** We conducted experiments on five real-world datasets of varying scales, including the academic coauthor networks DBLP (Hajiramezanali et al., 2019) and HepPh (Leskovec et al., 2005), the Ia-Enron employee email communication network (Rossi & Ahmed, 2015), the social media user communication network LFB (Viswanath et al., 2009), and the roll-call voting network in the United Nations General Assembly from 1946 to 2020 UNVote (Poursafaei et al., 2022). The statistics for these datasets are provided in Table 3. For each dataset except UNVote and Ia-Enron, we employed the same training and testing set splits as in prior work (Yang et al., 2021) across multiple snapshots. For UNVote and Ia-Enron datasets, we used a similar split ratio as applied to the other datasets. Additionally, we computed the global Gromov hyperbolicity $\delta$ for each dataset, which indicates that these datasets exhibit a certain degree of hyperbolicity.

| Scale | Datasets | # Links | # Entities | # Train / Test | $\delta$ |
|---|---|---|---|---|---|
| **Large** | UNVote | 1,035,742 | 201 | 72 / 6 | 0.5 |
| | HepPh | 976,097 | 9,746 | 8 / 3 | 1.0 |
| **Medium** | LFB | 180,011 | 45,435 | 33 / 3 | 2.0 |
| | Ia-Enron | 50,572 | 151 | 33 / 3 | 1.0 |
| **Small** | DBLP | 943 | 315 | 7 / 3 | 2.0 |

Table 3: Statistics of datasets.

| | Datasets | DBLP | | Ia-Enron | | LFB | | HepPh | | UNVote | |
|---|---|---|---|---|---|---|---|---|---|---|---|
| | Metrics | AUC | AP | AUC | AP | AUC | AP | AUC | AP | AUC | AP |
| Euclidean | EdgeBank | 19.00 ± 0.00 | 73.27 ± 0.00 | 20.41 ± 0.00 | 78.16 ± 0.00 | 13.31 ± 0.00 | 59.65 ± 0.00 | 15.68 ± 0.00 | 64.33 ± 0.00 | 22.11 ± 0.00 | 59.68 ± 0.00 |
| | GAE | 77.54 ± 0.33 | 74.21 ± 0.49 | 85.16 ± 0.78 | 87.06 ± 0.21 | 63.07 ± 0.93 | 65.35 ± 0.90 | 69.44 ± 0.56 | 73.61 ± 0.58 | 56.98 ± 0.19 | 60.40 ± 0.10 |
| | GRUGCN | 84.60 ± 0.92 | 87.87 ± 0.58 | 86.12 ± 0.10 | 87.51 ± 0.54 | 79.38 ± 1.02 | 82.77 ± 0.75 | 82.86 ± 0.53 | 85.87 ± 0.23 | 60.08 ± 0.27 | 60.75 ± 0.03 |
| | EvolveGCN | 83.88 ± 0.53 | 87.53 ± 0.22 | 83.34 ± 1.33 | 83.64 ± 1.33 | 76.85 ± 0.85 | 80.87 ± 0.64 | 76.82 ± 1.46 | 81.18 ± 0.89 | 58.74 ± 0.28 | 58.60 ± 0.15 |
| | DySAT | 87.25 ± 1.70 | 76.88 ± 0.08 | 78.16 ± 2.44 | 80.84 ± 1.77 | 76.88 ± 0.08 | 80.39 ± 0.14 | 81.02 ± 0.25 | 84.47 ± 0.23 | 59.43 ± 0.01 | 60.25 ± 0.05 |
| Hyperbolic | HGCN | 89.16 ± 0.16 | 91.63 ± 0.22 | 72.58 ± 0.84 | 72.37 ± 0.75 | 86.11 ± 0.13 | 83.74 ± 0.15 | 90.64 ± 0.07 | 88.98 ± 0.09 | 55.45 ± 0.14 | 58.54 ± 0.07 |
| | HAT | 89.29 ± 0.18 | 90.15 ± 0.14 | 75.84 ± 0.69 | 75.36 ± 0.65 | 84.02 ± 0.09 | 83.03 ± 0.15 | 90.52 ± 0.04 | 89.53 ± 0.04 | 55.33 ± 0.08 | 58.47 ± 0.05 |
| | HTGN-$\mathcal{B}$ | 89.26 ± 0.17 | 91.91 ± 0.07 | 85.37 ± 0.79 | 88.12 ± 0.70 | 83.70 ± 0.33 | 83.80 ± 0.43 | 91.13 ± 0.14 | 89.52 ± 0.28 | 61.40 ± 0.12 | 62.16 ± 0.10 |
| | HTGN-$\mathcal{L}$ | 88.56 ± 0.06 | 91.05 ± 0.11 | 85.14 ± 0.32 | 87.85 ± 0.41 | 82.98 ± 0.11 | 79.95 ± 1.26 | 91.33 ± 0.15 | 89.78 ± 0.15 | 61.27 ± 0.31 | 62.10 ± 0.32 |
| | HGWaveNet | 89.96 ± 0.27 | 92.12 ± 0.18 | 88.92 ± 0.70 | 90.70 ± 0.03 | 89.51 ± 0.28 | 86.88 ± 0.29 | 92.37 ± 0.04 | 91.48 ± 0.05 | 59.01 ± 0.03 | 60.32 ± 0.13 |
| | Ours | **94.13 ± 0.46** | **92.52 ± 0.47** | **94.17 ± 0.20** | **94.35 ± 0.50** | **93.31 ± 0.32** | **89.58 ± 0.41** | **97.19 ± 0.03** | **94.68 ± 0.05** | **61.68 ± 0.31** | **62.43 ± 0.06** |
| | Gain (%) | + 4.17 | + 0.40 | + 5.25 | + 3.65 | + 3.80 | + 2.70 | + 4.82 | + 3.20 | + 0.28 | + 0.27 |

Table 1: AUC (↑) and AP (↑) scores (%) of *temporal link prediction* on real-world dynamic graphs. For all methods, the best results are in **bold**, the suboptimal results are underlined, results within one standard deviation of best results are in shaded cells. Partial results are from (Bai et al., 2023) under the same experimental setup. These annotations also apply to Table 2.

| | Datasets | DBLP | | Ia-Enron | | LFB | | HepPh | | UNVote | |
|---|---|---|---|---|---|---|---|---|---|---|---|
| | Metrics | AUC | AP | AUC | AP | AUC | AP | AUC | AP | AUC | AP |
| Euclidean | EdgeBank | 77.73 ± 0.00 | 52.83 ± 0.00 | 14.42 ± 0.00 | 61.69 ± 0.00 | 11.36 ± 0.00 | 56.67 ± 0.00 | 14.24 ± 0.00 | 61.33 ± 0.00 | 9.55 ± 0.00 | 54.51 ± 0.00 |
| | GAE | 69.55 ± 0.57 | 66.11 ± 0.42 | 64.18 ± 4.63 | 66.68 ± 1.92 | 73.99 ± 0.21 | 77.56 ± 0.23 | 73.15 ± 0.18 | 79.57 ± 0.93 | 42.10 ± 1.84 | 52.78 ± 1.06 |
| | GRUGCN | 75.60 ± 1.60 | 78.55 ± 1.05 | 64.39 ± 1.74 | 67.91 ± 1.28 | 77.69 ± 1.03 | 81.07 ± 0.77 | 81.97 ± 0.49 | 84.78 ± 0.22 | 53.92 ± 3.78 | 61.19 ± 1.26 |
| | EvolveGCN | 73.49 ± 0.86 | 77.11 ± 0.44 | 67.45 ± 2.46 | 68.94 ± 0.90 | 74.49 ± 0.89 | 78.33 ± 0.66 | 74.79 ± 1.61 | 79.04 ± 1.02 | 28.49 ± 8.93 | 45.42 ± 5.46 |
| | DySAT | 79.74 ± 4.35 | 83.47 ± 3.01 | 63.80 ± 2.13 | 66.38 ± 1.45 | 74.97 ± 0.12 | 78.34 ± 0.07 | 79.01 ± 0.26 | 82.53 ± 0.25 | 68.42 ± 0.02 | 60.08 ± 0.03 |
| Hyperbolic | HGCN | 81.20 ± 0.19 | 83.28 ± 0.23 | 59.74 ± 1.05 | 59.33 ± 2.15 | 81.04 ± 0.14 | 80.59 ± 0.13 | 89.64 ± 0.27 | 87.87 ± 0.11 | 46.21 ± 3.46 | 36.96 ± 1.28 |
| | HAT | 79.29 ± 0.15 | 82.58 ± 0.08 | 61.34 ± 2.86 | 60.49 ± 0.34 | 83.05 ± 0.10 | 82.96 ± 0.18 | 89.63 ± 0.05 | 88.34 ± 0.04 | 47.13 ± 0.95 | 36.48 ± 0.25 |
| | HTGN-$\mathcal{B}$ | 81.74 ± 0.56 | 84.06 ± 0.41 | 62.93 ± 1.43 | 70.93 ± 0.76 | 82.21 ± 0.41 | 81.70 ± 0.46 | 90.11 ± 0.14 | 88.18 ± 0.31 | 50.15 ± 0.52 | 57.13 ± 3.60 |
| | HTGN-$\mathcal{L}$ | 81.53 ± 0.01 | 83.14 ± 0.22 | 61.73 ± 1.54 | 70.40 ± 0.08 | 81.67 ± 0.14 | 78.36 ± 1.03 | 90.34 ± 0.14 | 88.45 ± 0.14 | 52.10 ± 0.50 | 56.00 ± 0.60 |
| | HGWaveNet | 84.26 ± 0.46 | 86.19 ± 0.33 | 69.19 ± 0.17 | 74.01 ± 0.03 | 88.59 ± 0.28 | 86.00 ± 0.30 | 91.45 ± 0.05 | 90.21 ± 0.04 | 44.11 ± 4.87 | 52.37 ± 5.68 |
| | Ours | **93.95 ± 0.43** | **92.56 ± 0.42** | **89.68 ± 0.63** | **90.68 ± 0.64** | **93.25 ± 0.36** | **89.54 ± 0.47** | **97.18 ± 0.03** | **97.67 ± 0.07** | **68.50 ± 1.47** | **71.83 ± 2.40** |
| | Gain (%) | + 9.69 | + 6.37 | + 20.49 | + 16.67 | + 4.66 | + 3.54 | + 5.73 | + 7.46 | + 0.08 | + 10.64 |

Table 2: AUC (↑) and AP (↑) scores (%) of *temporal new link prediction* on real-world dynamic graphs.

**Baselines.** To comprehensively validate the superiority of our framework, we conducted extensive experiments using a variety of competitive baseline methods. For Euclidean space-based methods, we selected static graph model GAE (Kipf & Welling, 2016) and several high-performing dynamic graph models, including GRUGCN (Seo et al., 2018), EvolveGCN (Pareja et al., 2020), DySAT (Sankar et al., 2020), and EdgeBank (Yu et al., 2023), all of which exhibit strong performance in handling dynamic graph data. For hyperbolic space-based approaches, we chose representative static and dynamic graph models. Static graph models include HGCN (Chami et al., 2019) and HAT (Zhang et al., 2021), which are effective in hyperbolic space graph embedding. Dynamic graph models include HTGN-$\mathcal{B}$ (Yang et al., 2022), HTGN-$\mathcal{L}$ (Yang et al., 2022), and HGWaveNet (Bai et al., 2023), which offer effective solutions for dynamic graphs in hyperbolic space. By evaluating these baselines, we aim to highlight the relative advantages of our method and demonstrate its efficacy in processing dynamic graph data across different geometric spaces.

**Tasks and Metrics.** Our downstream tasks include *temporal link prediction* and *temporal new link prediction*. Specifically, given observed snapshots of a temporal graph $\mathcal{G} = \{\mathcal{G}_1, \ldots, \mathcal{G}_t\}$, the *temporal link prediction* task is de-

fined as predicting links in the next snapshot $\mathcal{G}_{t+1}$, and the *temporal new link prediction* task aims to identify new links in $\mathcal{G}_{t+1}$ that do not exist in $\mathcal{G}_t$. All methods employ the Fermi-Dirac function (Eq. 20) to perform these two link prediction tasks on the test sets of various datasets, and use AUC (Area Under the Curve) and AP (Average Precision) as evaluation metrics. To avoid errors due to randomness, each AUC and AP score is the average of five experimental results, with the standard deviation reported.

### 5.2. Experimental Results

**Temporal Link Prediction.** The results of temporal link prediction experiments for our method and all baseline methods across different-scale datasets are shown in Table 1. Our method outperforms all baseline methods in both AUC and AP metrics across all datasets, with particularly notable improvements on Ia-Enron (AUC +5.25%, AP +3.65%), LFB (AUC +3.80%, AP +2.70%), and HepPh (AUC +4.82%, AP +3.20%). These results demonstrate the significant advantage of our method in temporal link prediction, showing stability and reliability across datasets with various scales. Compared to the suboptimal results, our method improves AUC and AP by an average of 3.66% and 2.04%, respectively. While hyperbolic methods outperform

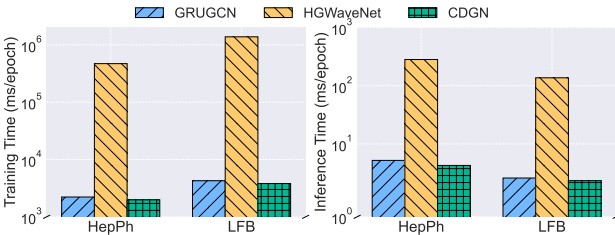

Figure 5: Efficiency comparison on `HepPh` and `LFB`.

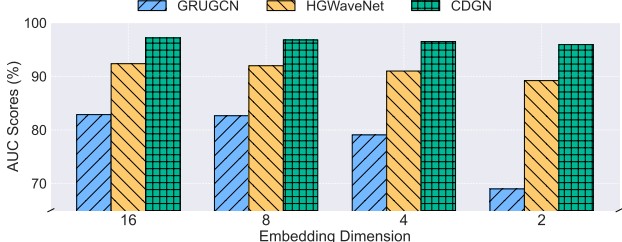

Figure 7: The influence of embedding dimension on `HepPh`.

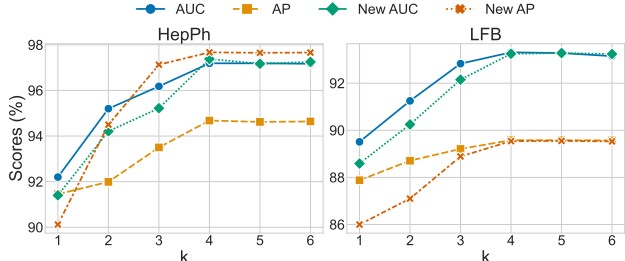

Figure 6: The influence of $k$ on `HepPh` and `LFB`.

| Tasks | *Temporal Link Prediction* | | *Temporal New Link Prediction* | |
|---|---|---|---|---|
| **Metrics** | **AUC** | **AP** | **AUC** | **AP** |
| **Full** | **93.31 $\pm$ 0.32** | **89.58 $\pm$ 0.41** | **93.25 $\pm$ 0.36** | **89.54 $\pm$ 0.47** |
| **w/o DGM** $\Delta(\%)$ | 90.89 $\pm$ 0.08 
 - 2.42 | 87.82 $\pm$ 0.14 
 - 1.76 | 90.48 $\pm$ 0.10 
 - 2.77 | 87.07 $\pm$ 0.06 
 - 2.47 |
| **w/o TSA** $\Delta(\%)$ | 92.10 $\pm$ 0.02 
 - 1.21 | 88.53 $\pm$ 0.05 
 - 1.05 | 92.16 $\pm$ 0.05 
 - 1.09 | 88.38 $\pm$ 0.11 
 - 1.16 |
| **w/o LEL** $\Delta(\%)$ | 91.77 $\pm$ 0.08 
 - 1.54 | 88.05 $\pm$ 0.04 
 - 1.53 | 88.40 $\pm$ 0.02 
 - 4.85 | 86.05 $\pm$ 0.05 
 - 3.49 |

Table 4: Ablation study.

Euclidean methods, they still fall short of our method. This is because our method effectively captures the geometric heterogeneity between different local structures in graph, thereby reducing representation distortion.

**Temporal New Link Prediction.** The results of temporal new link prediction experiments for all methods across different-scale datasets are shown in Table 2. These results demonstrate that our method outperforms all baseline methods on both AUC and AP metrics across all datasets. Notably, it achieves a 20.49% increase in AUC and a 16.67% increase in AP on the `Ia-Enron` dataset, a 9.69% increase in AUC on the `DBLP` dataset, and a 10.64% increase in AP on the `UNVote` dataset. On average, it improves AUC by 8.13% and AP by 8.94% compared to the suboptimal results. These substantial improvements validate the effectiveness of our proposed link evolution loss in capturing dynamic changes at a fine-grained link level, optimizing the embedding distribution to better align with the potential future distribution, and thus enhancing the performance of temporal new link prediction.

**Efficiency Comparison.** We compared the training and inference times of Euclidean SOTA GRUGCN, hyperbolic SOTA HGWaveNet and our method on `HepPh` and `LFB` datasets. As illustrated in Figure 5, our method achieves the highest computational efficiency. This is because, unlike HGWaveNet, which employs an expansion operation for historical information and embeds the graph data into the hyperbolic space at a coarse granularity, our method utilizes a straightforward aggregation layer to integrate historical information while leveraging complex hyperbolic

computations only for specific local structures.

**Ablation Study.** To validate the effectiveness and analyze the importance of key components in our method, we designed three ablation strategies on `HepPh`: (i) excluding DGM module (denoted as w/o DGM), using Euclidean embedding; (ii) excluding the TSA layer (denoted as w/o TSA), using the output representations from the previous timestamp as hidden state of next timestamp; (iii) excluding LEL (denoted as w/o LEL). The results of these ablation experiments conducted on `LFB` are presented in Table 4. The results indicate that the DGM module plays the most critical role, while the link evolution loss has the greatest impact on temporal new link prediction. Additionally, the temporal state aggregator layer also contributes to the improvement of model performance.

**Hyperparameter Analysis.** We study the impact of two hyperparameters in our method:

1) $k$ *of the $k$-hop ego-graph.* We varied the value of k in ego-graphs of the DGM module, ranging from 1 to 6. We conducted temporal link prediction and temporal new link prediction experiments on the `HepPh` and `LFB` datasets, respectively, with the results shown in Figure 6. The results indicate that when k is less than 4, both AUC and AP values are suboptimal. When k is 4 or greater, these metrics approach optimal values and stabilize. This suggests that a k value of 4 provides a comprehensive description of the local subgraph structure, consistent with the findings of (Adcock et al., 2013). Note that when the value of $k$ is too large, our method generally collapses into using only Euclidean space

| | Link Prediction Task | | | | New Link Prediction Task | | | |
| | DBLP | | la-Enron | | DBLP | | la-Enron | |
| | AUC | AP | AUC | AP | AUC | AP | AUC | AP |
|---|---|---|---|---|---|---|---|---|
| Clean | **93.11** | **92.32** | **91.43** | **89.17** | **91.23** | **90.14** | **89.59** | **87.16** |
| 5% (N) | 90.60 | 89.69 | 86.26 | 79.91 | 88.63 | 86.90 | 85.33 | 79.17 |
| 10% (N) | 88.49 | 89.50 | 83.24 | 76.87 | 86.82 | 86.89 | 81.69 | 74.83 |
| 20% (N) | 82.68 | 84.86 | 81.78 | 77.38 | 79.65 | 80.49 | 79.52 | 73.77 |
| 5% (M) | 92.16 | 89.79 | 87.91 | 81.17 | 89.57 | 86.26 | 88.26 | 81.84 |
| 10% (M) | 91.91 | 89.15 | 87.61 | 79.59 | 89.37 | 84.66 | 88.57 | 81.06 |
| 20% (M) | 92.25 | 88.17 | 89.19 | 80.43 | 89.36 | 83.30 | 89.06 | 80.80 |

Table 5: Performance under varying levels of noisy edges and missing edges on `DBLP` and `Ia-Enron` Datasets.

| | Link Prediction Task | | | | New Link Prediction Task | | | |
| | DBLP | | la-Enron | | DBLP | | la-Enron | |
| | AUC | AP | AUC | AP | AUC | AP | AUC | AP |
|---|---|---|---|---|---|---|---|---|
| Euclidean | 96.31 | 93.86 | 91.77 | 88.93 | 96.65 | 97.11 | 92.73 | 89.12 |
| Spherical | 96.60 | 93.49 | 92.40 | 88.62 | 96.30 | 96.72 | 92.95 | 88.97 |
| **Our** | **97.19** | **94.68** | **93.31** | **89.58** | **97.18** | **97.67** | **93.25** | **89.54** |

Table 6: Link prediction and new link prediction performance under different geometric embeddings.

embeddings (as most graphs are not purely tree-like), resulting in performance similar to the ablation setting without DGM, while still maintaining decent performance.

2) *Embedding dimension.* We varied the embedding dimensions to 2, 4, 8, and 16, and performed temporal link prediction experiments with our method, Euclidean SOTA GRUGCN, and hyperbolic SOTA HGWaveNet on the `HepPh` and `LFB` datasets. The results, illustrated in Figure 7, demonstrate that across different embedding dimensions, our method consistently outperforms both Euclidean and hyperbolic SOTA methods. Moreover, even when the embedding dimension is reduced to just 2, our method does not experience a significant drop in accuracy, unlike other methods. It is able to maintain a relatively high precision of approximately 95%, which demonstrates the strong capability of our approach in accurate modeling. This suggests that our method can effectively capture geometric information, even in lower-dimensional representation spaces. Optimal performance is achieved when embedding dimension is 16.

**Robustness Analysis.** To assess the robustness of our framework under more challenging conditions, we conducted additional experiments on `DBLP` and `Ia-Enron` datasets with two types of graph corruption: (1) Noisy edges (N), where we randomly inserted 5 %, 10 %, and 20 % of non-existing edges into the graph; (2) Missing edges (M), where we randomly deleted 5 %, 10 %, and 20 % of the original edges. The results, presented in Table 5, show that our method maintains high prediction accuracy across all corruption levels. Even when 20 % of edges are added or removed, the performance only degrades marginally. This demonstrates that our framework exhibits strong resistance to both noise and partial missingness in the underlying graph structure, confirming its robustness in imperfect real-world settings.

**Extension to Spherical Geometry.** Hyperbolic space is a complete, simply connected Riemannian manifold of constant negative curvature. To investigate whether positive curvature might benefit link prediction task, we replaced the hyperbolic component of our method with a spherical

space of constant curvature $c = +1.0$, while retaining the Euclidean component unchanged. Specifically, the entire graph is embedded either in Euclidean space ($c = 0.0$) or in spherical space, and experiments are re-run on `HepPh` and `LFB`. Performance is evaluated in AUC and AP, and results are presented in Table 6. The results indicate that the spherical embedding does not improve upon either the Euclidean baseline or our combined hyperbolic and Euclidean approach. We attribute this to the fact that spherical geometry is most effective when the data exhibit strong periodicity or closed-loop structures. Neither `HepPh` nor `LFB` possesses such global symmetries, so spherical embeddings fail to capture additional relational nuances. Thus, embedding geometry must align with graph structure.

## 6. Conclusion

In this paper, we propose a multi geometric dynamic framework to overcome the limitations of single geometric static or dynamic graph models. By combining Euclidean and hyperbolic geometries, our method effectively captures varying local structures. Additionally, the link evolution loss enhances predictive accuracy by modeling the intricate dynamics of link appearance and disappearance over time. Experiments show that our method significantly outperforms existing methods, including both Euclidean and hyperbolic models, offering a new paradigm for temporal link prediction through multi geometric perspective.

## Acknowledgments

This work was supported by National Natural Science Foundation of China, Grant Number: 62476109, 62206108, the Natural Science Foundation of Jilin Province, Grant Number: 20240101373JC, and Jilin Province Budgetary Capital Construction Fund Plan, Grant Number: 2024C008-5.

## Impact Statement

This paper presents work whose goal is to advance the field of Machine Learning. There are many potential societal consequences of our work, none which we feel must be specifically highlighted here.

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

## A. Notations

The descriptions of all the symbols used in the paper are provided in Table 7. In the order of appearance in the paper.

| Symbols | Descriptions |
|---|---|
| $\mathcal{G}$ | Discrete dynamic graph denoted by snapshots. |
| $\mathcal{V}$ | Entity set of discrete dynamic graph $\mathcal{G}$. |
| $\mathcal{E}$ | Link set of discrete dynamic graph $\mathcal{G}$. |
| $\mathbb{T}$ | Timestamp set of discrete dynamic graph $\mathcal{G}$. |
| $t$ | Current timestamp. |
| $\mathcal{G}_t$ | Snapshot of dynamic graph at timestamp $t$. |
| $\mathcal{E}_t$ | Link set of snapshot $\mathcal{G}_t$. |
| $t^+$ | Future timestamps relative to timestamp $t$. |
| $\Pr(\cdot)$ | Probability estimation function. |
| $\hat{y}_{uv}$ | Predicted 0-1 label indicating the existence of the link $(u, v)$. |
| $\hat{y}_{uv}^{absent}$ | Predicted 0-1 label indicating the non-existence of the link $(u, v)$. |
| $\mathbb{B}_c^n$ | $N$-dimensional Poincaré disk model with curvature $c$. |
| $g^{\mathbb{H}}$ | Riemannian metric of manifold $\mathbb{H}$. |
| $g^{\mathbb{B}}$ | Riemannian metric of manifold $\mathbb{B}$. |
| $g^{\mathbb{E}}$ | Euclidean norm. |
| $\mathcal{T}_{\boldsymbol{x}}\mathbb{B}_c^n$ | Tangent space at a point $\boldsymbol{x}$ in manifold $\mathbb{B}_c^n$. |
| $\delta_{i,t}$ | Hyperbolicity of entity $i$'s ego-graph at timestamp $t$. |
| $\ell(u_1, u_2)$ | Shortest path length between $u_1$ and $u_2$. |

| Symbols | Descriptions |
|---|---|
| $\mathbf{X}_t^{\mathbb{B}}$ | Embedding matrix in hyperbolic space modeled by poincaré disk model at timestamp $t$. |
| $\mathbf{X}_t^{\mathbb{E}}$ | Embedding matrix in Euclidean space at timestamp $t$. |
| $\mathbf{F}_{\delta_{i,t}\approx 0}$ | Feature matrix of the tree-like ego-graph centered of entity $i$ at timestamp $t$. |
| $\mathbf{F}_{\delta_{i,t}\not\approx 0}$ | Feature matrix of the non-tree-like ego-graph centered of entity $i$ at timestamp $t$. |
| $\sigma^{\otimes^c}$ | Geometric non-linear activation with different curvatures $c$. |
| $\phi(\cdot)$ | Nonlinear activation function such as hyperbolic tangent function. |
| $\gamma(\cdot)$ | Scalar function with default setting of $f(x) = x$. |
| $\alpha$ | Attention vector corresponding to the set of past timestamps. |
| $\mathcal{E}_{new}$ | Link set satisfying $\mathcal{E}_{\text{new}} \subseteq \mathcal{E}_t$ and $\mathcal{E}_{\text{new}} \cap \mathcal{E}_{t-1} = \emptyset$ |
| $\mathcal{E}_{lost}$ | Link set satisfying $\mathcal{E}_{\text{lost}} \subseteq \mathcal{E}_{t-1}$ and $\mathcal{E}_{\text{lost}} \cap \mathcal{E}_t = \emptyset$ |
| $|\cdot|$ | Absolute value function. |
| $\tau$ | Penalty term, which is typically set to 0.5. |

Table 7: Notations.

## B. Background of Geometry

### B.1. Euclidean Geometry

Euclidean geometry is a fundamental branch of mathematics that examines the properties and relationships of shapes and spaces with zero curvature. This geometry is based on a flat, infinite plane where parallel lines never converge, and the sum of the angles in a triangle always equals 180 degrees. It forms the basis for most classical and contemporary mathematical modeling and neural network operations.

In Euclidean space, volume grows polynomially with the growth of radius, and the majority of neural network models are designed to operate within this space. Operations such as convolution, pooling, and activation functions are performed using basic arithmetic operations like addition, subtraction, multiplication, and division.

### B.2. Hyperbolic Geometry

Hyperbolic geometry represents a non-Euclidean geometric model with constant negative curvature, contrasting sharply with the zero curvature of Euclidean space. The negative curvature in hyperbolic space leads to exponential growth of volume with respect to the growth of radius, which contrasts with the polynomial growth in Euclidean space. This characteristic allows hyperbolic models to efficiently handle data with hierarchical or complex structures, which are difficult to represent in Euclidean space.

Hyperbolic space can be modeled using several isomorphic representations, including the Lorentz model, Poincaré disk model, Poincaré half-space model, and Klein model. In our study, we employ the Poincaré disk model due to its advantageous

| Parameters | DBLP | Ia-Enron | LFB | HepPh | UNVote | UNLegis |
|---|---|---|---|---|---|---|
| # Layers of Encoder | 3 | 4 | 3 | 3 | 3 | 3 |
| # Snapshots of Test | 3 | 3 | 3 | 3 | 6 | 3 |
| Learning Rate | 0.0005 | 0.001 | 0.005 | 0.001 | 0.01 | 0.01 |
| Weight Decay | 0.0000005 | 0.0000005 | 0.0000005 | 0.0000005 | 0.0000005 | 0.0 |
| Hidden Layer Dimension | 16 | 16 | 16 | 16 | 16 | 16 |
| Dropout Rate | 0.0 | 0.0 | 0.0 | 0.0 | 0.1 | 0.0 |
| Initial Curvature Setting | None | None | None | None | None | None |
| $r$ in Fermi-Dirac Decoder | 2.0 | 2.0 | 2.0 | 2.0 | 2.0 | 2.0 |
| $t$ in Fermi-Dirac Decoder | 1.0 | 1.0 | 1.0 | 1.0 | 1.0 | 1.0 |

Table 8: Parameter settings.

properties for graph-based data. The Poincaré disk model $\mathbb{B}^n$ is defined as:

$$\mathbb{B}^n = \{x \in \mathbb{R}^n : \|x\| < 1\},$$

where $\| \cdot \|$ denotes the Euclidean norm.

In this model, the distance between points $x$ and $y$ is given by:

$$d(x, y) = \operatorname{arcosh} \left( 1 + 2 \frac{\|x - y\|^2}{(1 - \|x\|^2)(1 - \|y\|^2)} \right).$$

Key operations in hyperbolic geometry include:

- **Möbius Addition** $\oplus$:

$$x \oplus y = \frac{\left(1 + 2\langle x, y \rangle + \|y\|^2\right) x + \left(1 - \|x\|^2\right) y}{1 + 2\langle x, y \rangle + \|x\|^2 \|y\|^2}.$$

- **Möbius Scalar Multiplication** $\otimes$:

$$r \otimes x = \begin{cases} \tanh\left(r \operatorname{artanh}(\|x\|) \frac{x}{\|x\|}\right), & \text{if } x \in \mathbb{B}^n, \\ 0, & \text{if } x = 0. \end{cases}$$

- **Möbius Vector Multiplication** $M^{\otimes}(x)$:

$$M^{\otimes}(x) = \tanh\left(\frac{\|Mx\|}{\|x\|} \operatorname{actanh}(\|x\|)\right) \frac{Mx}{\|Mx\|}.$$

These operations leverage the hyperbolic space's properties to handle complex relationships and structures, which are not easily manageable in Euclidean space. Our use of the Poincaré disk model aims to capture these intricate structures effectively in our experiments.

| | Tasks | *Temporal Link Prediction* | | Temporal New Link Prediction | |
|---|---|---|---|---|---|
| | Metrics | AUC | AP | AUC | AP |
| Euclidean | EdgeBank | $65.78 \pm 0.00$ | $16.30 \pm 0.00$ | $\underline{50.04 \pm 0.00}$ | $19.05 \pm 0.00$ |
| | GAE | $50.62 \pm 0.53$ | $51.81 \pm 0.09$ | $34.17 \pm 1.59$ | $39.97 \pm 0.50$ |
| | GRUGCN | $53.26 \pm 0.15$ | $56.39 \pm 0.17$ | $33.08 \pm 0.25$ | $38.89 \pm 0.45$ |
| | EvolveGCN | $57.55 \pm 0.21$ | $\underline{58.62 \pm 0.71}$ | $39.27 \pm 0.14$ | $41.48 \pm 0.38$ |
| | DySAT | $53.95 \pm 0.28$ | $56.69 \pm 0.11$ | $34.24 \pm 0.39$ | $39.76 \pm 0.17$ |
| Hyperbolic | HGCN | $51.61 \pm 0.51$ | $52.83 \pm 0.45$ | $37.36 \pm 3.85$ | $42.14 \pm 2.75$ |
| | HAT | $53.70 \pm 1.24$ | $54.87 \pm 1.08$ | $39.31 \pm 4.61$ | $43.79 \pm 0.39$ |
| | HTGN-$\mathcal{B}$ | $52.89 \pm 0.50$ | $52.88 \pm 0.37$ | $40.51 \pm 0.01$ | $\underline{43.89 \pm 1.01}$ |
| | HTGN-$\mathcal{L}$ | $\underline{57.87 \pm 0.71}$ | $56.89 \pm 0.38$ | $39.91 \pm 1.52$ | $42.09 \pm 1.22$ |
| | HGWaveNet | $55.63 \pm 0.22$ | $56.99 \pm 0.51$ | $32.55 \pm 0.86$ | $38.42 \pm 0.25$ |
| | Ours | $\mathbf{80.07 \pm 0.47}$ | $\mathbf{71.93 \pm 0.35}$ | $\mathbf{74.56 \pm 0.41}$ | $\mathbf{68.59 \pm 0.44}$ |
| | Gain (%) | + 22.2 | + 13.31 | + 24.52 | + 24.7 |

Table 9: AUC (↑) and AP (↑) scores (%) of *temporal link prediction* and *temporal new link prediction* on `USLegis` graphs. For all methods, the best results are in **bold**, the suboptimal results are underlined.

## C. Experimental Details

---
**Algorithm 1** Cross-Geometric Dynamic Link Prediction Framework

---
**Input**: Graph snapshots $\{\mathcal{G}_t\}_{t=0}^{T}$, where $\mathcal{G}_t = (\mathcal{V}_t, \mathcal{E}_t)$
**Parameter**: Initial hidden state $\mathbf{H}_0$, number of hops $k$ for ego-graph
**Output**: Final graph representation $\mathbf{H}_T$

1: **for** each timestamp $t = 0$ to $T - 1$ **do**
2:     **Dynamic Geometric Modeling:**
3:     **for** each node $v_i \in \mathcal{V}_t$ **do**
4:         Compute $k$-hop ego-graph $\mathcal{G}_{i,t}^{(k)}$
5:         Calculate hyperbolicity $\delta_{i,t}$ of $\mathcal{G}_{i,t}^{(k)}$
6:         **if** $\delta_{i,t} \approx 0$ **then**
7:             Map features $\mathbf{F}_{i,t}^{\mathbb{E}}$ into hyperbolic space
8:             Generate hyperbolic embedding $\mathbf{F}_{i,t}^{\mathbb{B}}$
9:         **else**
10:            Generate Euclidean embedding $\mathbf{F}_{i,t}^{\mathbb{E}}$
11:         **end if**
12:     **end for**
13:     Aggregate cross-geometric embeddings to form $\mathbf{X}_t$
14:     **Temporal State Aggregator:**
15:     Map historical representations $\{\mathbf{Z}_0, \mathbf{Z}_1, \ldots, \mathbf{Z}_{t-1}\}$ to higher-dimensional space
16:     Compute attention scores $\alpha_i$ for $i = 0$ to $t - 1$
17:     Predict current hidden state $\hat{\mathbf{H}}_t = \sum_{i=0}^{t-1} \alpha_i \cdot \mathbf{Z}_i$
18:     Identify edge sets $\mathcal{E}_{\text{new}}$ and $\mathcal{E}_{\text{lost}}$
19:     Compute loss $\mathcal{L}_t$ based on $\mathcal{E}_{\text{new}}$ and $\mathcal{E}_{\text{lost}}$
20:     **Optimization:**
21:     Update model parameters using gradient descent:
22:     $\theta \leftarrow \theta - \eta \cdot \nabla_\theta(\mathcal{L}_{\text{LEL},t} + \mathcal{L}_{\text{LP},t})$
23: **end for**
24: **return** Final graph representation $\mathbf{H}_T$

---

### C.1. Algorithm

To clearly illustrate the workflow of our proposed method, we provide a detailed description of its core steps here. Initially, for each timestamp, we extract the $k$-hop ego graph for each node from the given graph snapshot and compute its hyperbolicity

to determine the appropriate embedding space. Subsequently, by leveraging cross-geometric embeddings and temporal state aggregation, we generate the node representations for the current timestamp. Finally, we optimize the link evolution loss function to enhance the accuracy of dynamic link prediction. The detailed pseudocode is provided in Algorithm 1.

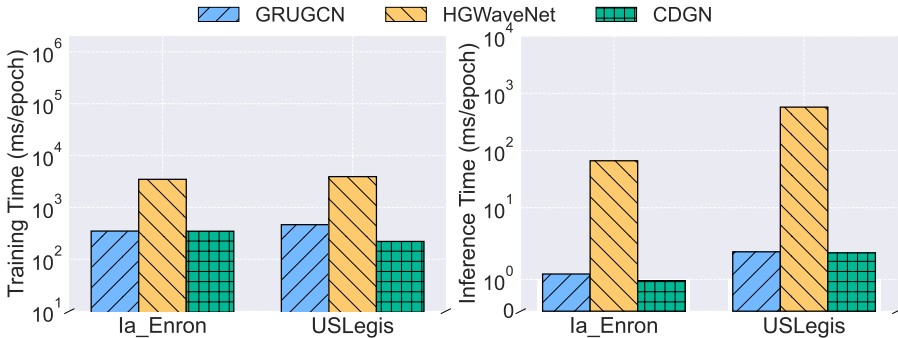

Figure 8: Efficiency Comparison on `Ia-Enron` and `USLegis`.

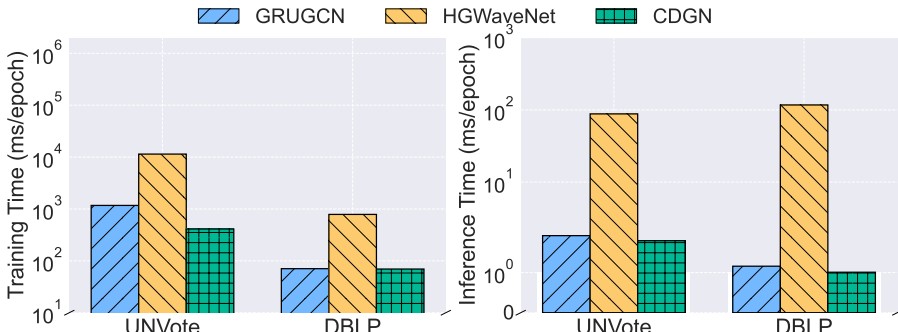

Figure 9: Efficiency Comparison on `UNVote` and `DBLP`.

### C.2. Data Preprocessing

Partial datasets are from (Yang et al., 2021), and thus the same dataset preprocessing methods are employed. The specific details are as follows:

`LFB` (Viswanath et al., 2009) is a social network graph of Facebook Wall posts where each entity is a user and each link is the interaction related to their wall posts. We take the activates over the last three years in the dataset as 36 snapshots. The FB dataset is associated with a large number of users but very sparse connections.

`HepPh` (Leskovec et al., 2005) is a citation network related to high energy physics phenomenology, which is collected from the e-print arXiv website. Each entity represents a paper, and an link represents one paper citing another. The data covers papers in the period between January 1993 to April 2003 (124 months in total). It is a directed graph network, but we learn and predict as if it was an undirected graph. According to the real physical meaning, we use three months of data per snapshot and use the last 36 months as the full dataset in our work.

`DBLP` (Hajiramezanali et al., 2019) is an academic cooperation network, including the academic cooperation of 315 researchers from 2000 to 2009. Each entity on the graph represents an author, and an link denotes a co-authorship relation. We split the dataset by year and obtain 10 snapshots.

The remaining datasets are processed using similar preprocessing methods. The details are as follows:

`UNVote` (Poursafaei et al., 2022) is a dataset of roll-call votes in the United Nations General Assembly from 1946 to 2020. Each entity represents a nation. If two nations both voted "yes" for an item, then the link weight between them is incremented by one. We split the dataset by year and obtain 78 snapshots.

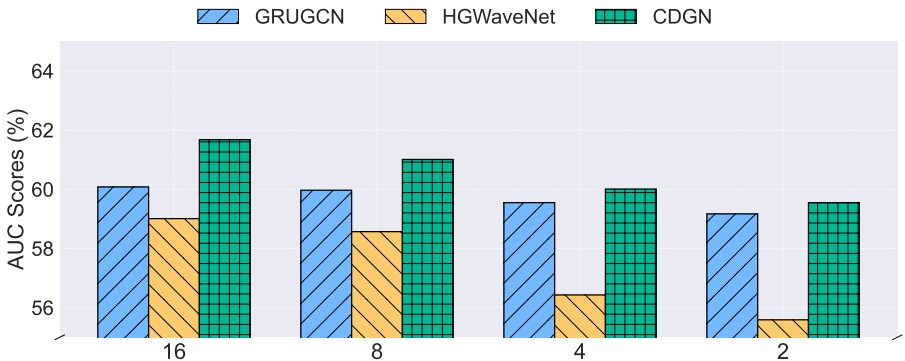

Figure 10: The influence of embedding dimension on `UNVote`.

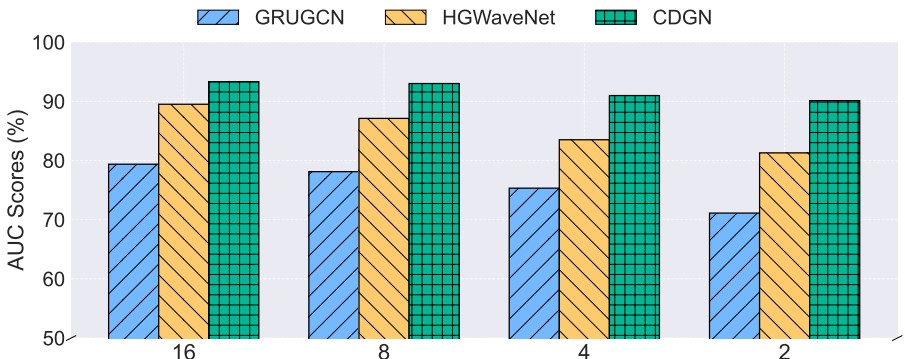

Figure 11: The influence of embedding dimension on `LFB`.

`Ia-Enron` (Rossi & Ahmed, 2015) is a dataset of edges that represent emails sent from one employee to another. There are 50572 links, and each of them contains timestamp information. Links refer to 151 unique entity IDs in total. The dataset comprises link data over 38 months. However, since the records for the first and last months are incomplete, they are merged with the second month and the second-to-last month, respectively. Consequently, the dataset is partitioned into 36 snapshots based on the months.

In the appendix, we additionally present the comparative experimental results of our method on a new dataset. The details of this dataset are as follows:

`USLegis` (Poursafaei et al., 2022) is a senate co-sponsorship graph which documents social interactions between legislators from the US Senate. The link weights specify the number of times two congress persons have co-sponsored a bill in a given congress. The dataset contains 12 months of data, comprising 225 entities and 60,396 links. It is divided into 12 snapshots based on the months, and split into training and test sets with a 9:3 ratio.

### C.3. Environments

The hardware environment consists of an Intel Core i7-13700KF CPU with 16 cores and 24 threads, running at 3.40GHz, paired with an NVIDIA GeForce RTX 4070Ti GPU that has 12GB of VRAM and 7680 CUDA cores. The system is equipped with 16GB of RAM and operates on Windows 11. Programming is performed using Python 3.10, with PyTorch 1.13.1 and torch geometric 2.2.0 for deep learning, CUDA 11.7 for GPU acceleration, and package management handled by Anaconda 3.0. For processing large datasets like `UNVote` and `HepPh`, a high-performance server is utilized, featuring 4 Intel Xeon Gold 5220 CPUs, each with 18 cores and 36 threads, clocked at 2.20GHz. This server also includes 4 NVIDIA Quadro RTX 6000 GPUs, each with 24GB of VRAM and 4608 CUDA cores, and is equipped with 500GB of RAM. The server runs on Ubuntu 18.04.6.

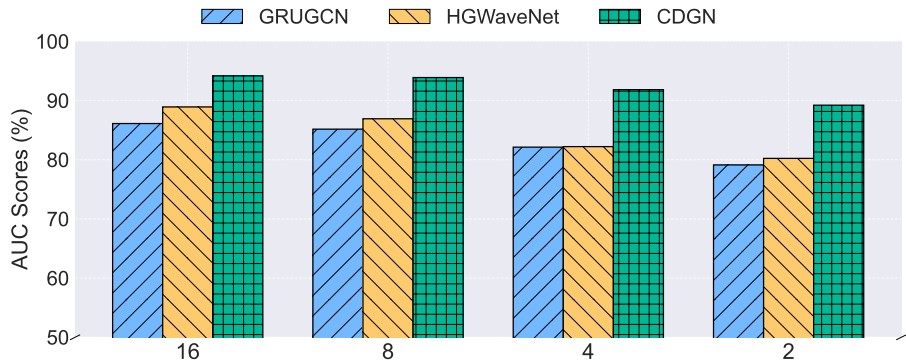

Figure 12: The influence of embedding dimension on `Ia-Enron`.

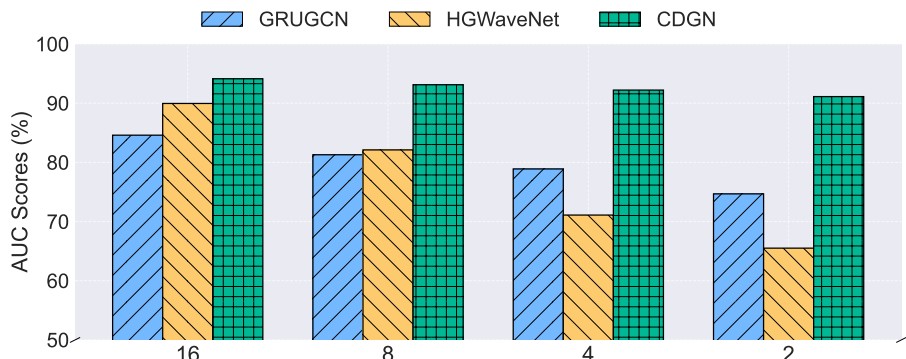

Figure 13: The influence of embedding dimension on `DBLP`.

### C.4. Parameter Settings

Our method primarily depends on the following hyperparameters: hidden layer dimension, number of test set snapshots, learning rate, weight decay, number of encoder layers, dropout rate, initial curvature setting (where *None* indicates that it is trainable), and the parameters $r$ and $t$ for the Fermi-Dirac decoder. The parameter settings used for different datasets are detailed in Table 8.

## D. Supplementary Experiments

### D.1. Further Performance Comparison

We conducted experiments on a new dataset, comparing the performance of various hyperbolic and Euclidean dynamic graph models with our method on both temporal link prediction and temporal new link prediction tasks. The results, as shown in Table 9, demonstrate that our method consistently outperforms all baseline methods. Specifically, it achieves an AUC score improvement of 22.2% and an AP score improvement of 13.31% on the temporal link prediction task, while on the temporal new link prediction task, the AUC score increases by 24.52% and the AP score increases by 24.7%.

### D.2. Further Efficiency Comparison

Due to space limitations, only a subset of the dataset efficiency comparisons is presented in the main paper. We include the remaining efficiency comparisons for other datasets here. Efficiency comparisons on the `Ia-Enron` and `USLegis` datasets are shown in Figure 8, while comparisons on the `UNVote` and `DBLP` datasets are illustrated in Figure 9. The results indicate that our method consistently demonstrates the best efficiency on `USLegis` datasets.

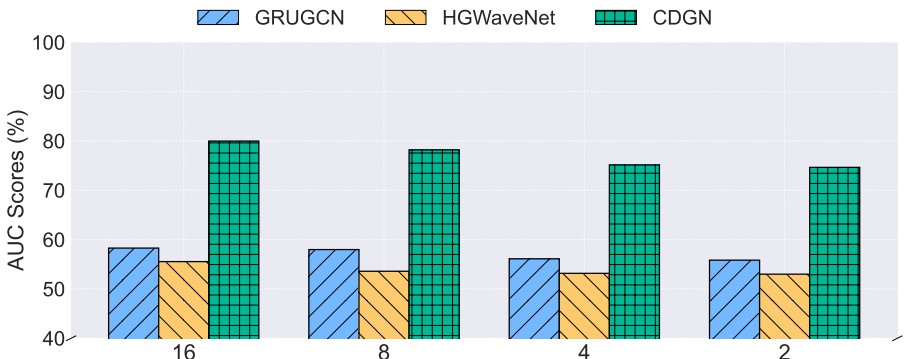

Figure 14: The influence of embedding dimension on `USLegis`.

### D.3. Further Hyperparameter Analysis

Due to space constraints, the main paper only presents the analysis of the hyperparameter embedding dimension on a subset of datasets. We provide the remaining analyses here. The analysis of embedding dimension on the UNVote dataset is shown in Figure 10, on the LFB dataset in Figure 11, on the Ia-Enron dataset in Figure 12, on the DBLP dataset in Figure 13, and on the USLegis dataset in Figure 14. The results indicate that the embedding dimension has minimal impact on the performance of our mehtod, with the best performance observed at a dimension of 16. Additionally, it outperforms both the hyperbolic SOTA method HGWaveNet and the Euclidean SOTA method GRUGCN across all dimension values.

