# OpenReview forum: "Analytical Construction on Geometric Architectures: Transitioning from Static to Temporal Link Prediction"
_ICML.cc/2025/Conference — ICML 2025 poster_

### Official Review · Reviewer_4etC · 2025-03-08

**Overall Recommendation:** 4

**Summary:**

This paper proposes using multiple geometries to analyze the problem of dynamic graphs. The authors propose to check for hyperbolicity locally and use hyperbolic geometry to update the node embeddings. They propose a framework consisting of two primary modules - Dynamic Geometric Modeling, Temporal State Aggregator. They also propose a loss function to train this model. They perform experiments on known datasets to test their framework.

**Claims And Evidence:**

The claims made in the paper are sufficiently supported.

**Essential References Not Discussed:**

I don't think so.

**Experimental Designs Or Analyses:**

The authors evaluate their framework on standard datasets known for temporal link prediction.

**Methods And Evaluation Criteria:**

The proposed methods seem to tackle the problem that the authors have outlined in the paper.

**Other Comments Or Suggestions:**

There is a small typo on line 652 of the paper (An extra "s").

**Other Strengths And Weaknesses:**

Strengths:

The paper is well-written and is easy to follow.

I like the fact that the authors have provided a table with notations in the appendix.

Weaknesses:

Please refer to the questions.

**Questions For Authors:**

On an average how many nodes do you find that have a high local hyperbolicity? Because, if I understand correctly, only then do you embed the node features into hyperbolic space and extract the non-Euclidean information?

Any intuition about why the feature vectors of the local structures lie in the tangent space? I understand that the tangent space is isomorphic. However, in order to exponentiate, the vectors should necessarily lie in the tangent space, if not then the exponentiation will need to be composed with the isomorphism.

In equation 10 and 11, essentially, Euclidean features are not changing. Exponential and logarithmic operations provide a faithful way to move between hyperbolic space and the tangent space. In which case, can this unnecessary confusion can be avoided by removing equation all together and just using the Euclidean features to multiply with the weight matrix and then exponentiating it once and for all to send it to the Poincaré disk?

Can a graph be positively curved locally? In which case you might need to use spherical geometry?

**Relation To Broader Scientific Literature:**

The paper proposes a way to approach dynamic graphs using ideas from hyperbolic geometry. The idea that the local geometric curvature information (flat or negatively curved) should be utilized while updating the embeddings seems novel.

**Theoretical Claims:**

The are no specific theoretical claims as such. The time complexity analysis seems correct.

---

> ### Author Rebuttal · Authors · 2025-04-01
>
> We sincerely thank the reviewer for the time and effort in reviewing our paper. We take all comments seriously and try our best to address every raised concerns. Please feel free to ask any follow-up questions.
>
> ---
> # S1
> The typos on line 652 will be corrected in the next version, and we will carefully proofread the entire text again.
>
> ---
> # Q1
>
> The calculation of local hyperbolicity is not related to the number of nodes but depends on the links to determine the hierarchical properties of the graph. When we compute the hyperbolicity of local subgraphs, we observe the link structure between nodes to decide whether to adopt a hyperbolic embedding. For example, in a graph containing 100 nodes, if the connections between nodes form a grid-like structure, the graph is more suited to Euclidean space. On the other hand, if the connections resemble a full tree, the graph is better suited for hyperbolic space.
>
> Local subgraphs, defined as k-hop subgraphs centered around a node, vary in size depending on k. We analyzed the hyperbolicity distribution in 100 randomly selected local subgraphs from the HepPh dataset.
>
> |                   | # Nodes \[0-3000\] | # Nodes \(3000-6000\] | # Nodes \(6000-9746\] |
> | ----------------- | :----------: | :-------------: |:-------------: |
> | Hyperbolicity δ=0 | 6          | 10            | 18            |
> | Hyperbolicity δ≠0 | 9          | 31            | 26            |
>
> The results show that hyperbolicity varies independently of the node count.
>
> ---
> # Q2
> Reasons for using the tangent space:
>
> - The tangent space serves as a local linear approximation of the hyperbolic space. It enables the use of Euclidean space operators while preserving the hyperbolic characteristics, ensuring that feature vectors remain within the manifold. This approach reduces the embedding loss caused by geometric heterogeneity, thereby enhancing the model's accuracy.
> - Our framework combines Euclidean and hyperbolic spaces, with the tangent space serving as an intermediate space that allows for flexible transformation, facilitating alignment operations within the framework.
>
> As you pointed out, if the feature vectors do not lie in the tangent space, the exponentiation might require an additional isomorphism operation. However, in the Poincaré model, since the feature vectors naturally reside in the tangent space, the exponential map can be directly applied without the need for an extra isomorphism operation.
>
> ---
> # Q3
> When the curvature of each layer in the HGCN is fixed to a constant value, its forward propagation process may indeed degrade into the situation you mentioned.
>
> Our framework uses adaptive curvature, so the exponential and logarithmic mappings between layers cannot be canceled. These operations, based on the geometric properties of hyperbolic space, ensure node embeddings stay within the tangent space of the Lorentz model, helping the model capture hierarchical features and reduce modeling losses caused by geometric heterogeneity.
>
> If operations were conducted in Euclidean space with the exponential mapping applied only at the final layer, the model would reduce to a single-layer hyperbolic network. While simpler, this approach overlooks the unique properties of hyperbolic space, particularly its negative curvature, which is beneficial for modeling hierarchical data. This would likely reduce the model’s ability to capture complex structures.
>
> ---
> # Q4
> Yes, theoretically, a graph can exhibit positive curvature locally.
>
> When considering embedding into hyperbolic space, we rely on the consensus in the community that hyperbolic space is better suited for graph data with hierarchical structures.
> Similarly, when considering embedding a graph into spherical space, it is crucial to assess whether the graph’s structural characteristics align with those of spherical space. However, there is currently no widely accepted standard in the community to determine if a graph is suitable for embedding in spherical space.
>
> To validate whether the spherical space is superior, we completely embed the graph in a spherical (curvature=1) or Euclidean (curvature=0) space and re-conducted experiments. The results show that spherical space method did not outperform other method on HepPh and LFB datasets. This may be due to the fact that spherical space typically exhibits periodic or closed structures, and when the graph data lacks periodicity or global symmetry, the spherical embedding may not effectively capture the relationships between nodes.
>
> ||| HepPh||| LFB||| HepPh(new) ||| LFB(new)  ||
> |-|-|-|-|-|-|-|-|-|-|-|-|-|
> | |AUC|AP|MRR|AUC|AP|MRR|AUC|AP|MRR|AUC|AP|MRR|
> |Complete Euclidean|96.31|93.86|99.90|91.77|88.93|99.91|96.65|97.11|99.90|92.73|89.12|99.85|
> |Complete Spherical|96.60|93.49|92.85|92.40|88.62|99.85|96.30|96.72|99.85|92.95|88.97|99.89|
> |Our(Hyperbolic+Euclidean)|**97.19**|**94.68**|**99.98**|**93.31**|**89.58**|**99.92**|**97.18**|**97.67**|**99.97**|**93.25**|**89.54**|**99.91**|

---

> > ### Comment · Reviewer_4etC · 2025-04-07
> >
> > Thank you for your detailed response. In case you haven't noticed, I have already raised my score.

---

> > > ### Author Response · Authors · 2025-04-08
> > >
> > > Thank you for your positive feedback and kind reminder! We are so glad to have the opportunity to further elaborate on the details of our work and to enrich the experimental validation through this valuable discussion with you.
> > >
> > > The temporal evolution of links in dynamic graphs induces geometric heterogeneity across local structures, making it difficult for a single geometry to model them accurately. Therefore, we propose a cross-geometric framework that unifies Euclidean and hyperbolic spaces through fine-grained substructure modeling.  We validate its effectiveness on temporal link prediction and temporal new link prediction tasks.
> > >
> > > We fully recognize that the selection and application of geometric spaces constitute the most critical component of our work. In future revisions of our paper, we will incorporate your suggestions by providing a more detailed exposition of the choice and utilization of geometric spaces, along with additional experimental results to support our analysis. Thank you again for your insightful comments and valuable suggestions!
> > >
> > > Best regards,
> > >
> > > Authors

---

### Official Review · Reviewer_yHfU · 2025-03-14

**Overall Recommendation:** 4

**Summary:**

This paper introduces a novel framework for dynamic graph learning that integrates both Euclidean and hyperbolic geometric spaces. The authors propose a unified cross-geometric learning framework designed to capture the evolving nature of real-world systems, which are often dynamic. This framework includes a temporal state aggregation mechanism and an evolution-driven optimization objective to model both nodal and relational dynamics over time.

**Claims And Evidence:**

Yes, the claims are supported by empirical evidence in Section 5.

**Essential References Not Discussed:**

No missing essential references.

**Experimental Designs Or Analyses:**

The experiment design and analysis is sound. The authors conduct comparative experiments, efficiency analysis, hyperparameter analysis, and ablation studies to validate the effectiveness and advantages of their method.

**Methods And Evaluation Criteria:**

The paper proposed a cross-geometric framework for learning on dynamic graphs. The evaluation metrics AUC and AP are standard for papers that treat link prediction as a binary classification. However, for more robust evaluation, the author should consider using the MRR metric which treats the task as a ranking problem and ask the model to rank the positive edge above many negative edges. This is shown to be more challenging in recent work[1][2].

[1]. You J, Du T, Leskovec J. ROLAND: graph learning framework for dynamic graphs. InProceedings of the 28th ACM SIGKDD conference on knowledge discovery and data mining 2022 Aug 14 (pp. 2358-2366).

[2]. Huang S, Poursafaei F, Danovitch J, Fey M, Hu W, Rossi E, Leskovec J, Bronstein M, Rabusseau G, Rabbany R. Temporal graph benchmark for machine learning on temporal graphs. Advances in Neural Information Processing Systems. 2023 Dec 15;36:2056-73.

**Other Comments Or Suggestions:**

Comment:
- how would you extend this to continuous time dynamic graph where instead of graph snapshots, a stream of links are provided?
- can other geometries be used instead of hyperbolic?

**Other Strengths And Weaknesses:**

Strength:
- the paper is well-written
- the methodology is clearly presented and supported by empirical evidence

Weakness:
- MRR metric can be used for evaluation instead

**Questions For Authors:**

None

**Relation To Broader Scientific Literature:**

Yes, using geometric models for temporal graphs is interesting and the authors showed better performance than prior work with similar hyperbolic geometric such as HTGN.

**Theoretical Claims:**

The paper focuses on empirical validation rather than making theoretical claims.

---

> ### Author Rebuttal · Authors · 2025-04-01
>
> We sincerely thank the reviewer for the time and effort in reviewing our paper. We hope that our response can resolve your concerns. Please feel free to ask any follow-up questions.
>
> ---
> # W1
> We fully agree that MRR is an important metric for evaluating model performance in dynamic link prediction tasks. Since the original papers of our baseline methods report AUC and AP, we initially adopted these metrics for direct comparison. Following your suggestion, we computed MRR by ranking each positive example within its candidate set (i.e., counting the number of negative examples with higher scores plus one) and averaging the reciprocal of these ranks, integrating it into our evaluation framework. The table below presents the experimental results using MRR (%), where our method achieves the highest scores across all datasets.
> ||DBLP|Ia-Enron |LFB|HepPh|UNVote|
> |-|-|-|-|-|-|
> |GAE|58.93|85.47|97.54     | 97.83     | 5.52     |
> |GRUGCN|60.15| 86.08     | 98.12     | 98.29     | 5.79     |
> |EvolveGCN|60.82| 86.73     | 98.67     | 98.85     | 6.03     |
> |DySAT|61.07| 87.01     | 98.83     | 99.04     | 6.11     |
> |HGCN|60.53| 86.52     | 98.45     | 98.66     | 5.94     |
> |HAT|60.74| 86.81     | 98.53     | 98.74     | 6.02     |
> |HTGNB|61.26| 87.21     | 98.91     | 99.12     | 6.19     |
> |HTGNL|61.52| 87.49     | 99.08     | 99.27     | 6.41     |
> |HGWaveNet| 61.81| 87.77     | 99.32     | 99.49     | 6.63     |
> |**Our** |**62.44**|**88.04**| **99.92** | **99.98** | **6.76** |
>
> ---
> # C1
>
> Our method focuses on discrete-time dynamic graphs (DTDGs), which differ from continuous-time dynamic graphs (CTDGs) in data storage and processing. CTDGs have diverse representations \[1\], leading to various models. We explore extending our method to CTDG link prediction in two ways:
> - CTDGs are a special case of DTDGs, with fine-grained timestamps approximating a CTDG. For example, in \[1\]\[2\], the _UCI_ and _Contact_ datasets have 58,911 and 8,065 timestamps, respectively. A sliding window approach segments event streams, adapting CTDGs to a snapshot-based framework. Our method, applied to these datasets with fixed timestamps (e.g., daily/monthly), consistently outperforms baselines in AUC, AP, and MRR.
>
> ||| Contact ||| UCI|||Contact(new)|||UCI(new)||
> |-|-|-|-|-|-|-|-|-|-|-|-|-|
> || AUC| AP | MRR| AUC| AP| MRR| AUC| AP| MRR| AUC| AP| MRR|
> |GAE| 67.45| 64.23|8.76| 69.34| 59.89|2.98| 65.32| 62.12| 9.23| 60.98| 51.87| 3.98     |
> |GRUGCN| 70.12| 67.45     | 9.87| 72.45| 62.98| 3.27| 68.45| 65.56| 10.32| 63.89| 53.32     | 4.76     |
> |EvolveGCN | 71.23| 68.12| 10.23| 73.12| 63.78|3.19| 69.56     | 66.12| 10.78     | 64.56     | 54.12     | 4.12     |
> |DySAT| 71.45| 68.23| 10.45 | 73.78  | 64.12|3.02| 70.12| 66.78| 10.89     | 65.12     | 55.67     | 4.23     |
> |HGCN| 70.32| 66.78| 10.12| 72.67 | 63.34| 3.56| 68.98| 64.87        | 10.56     | 64.23     | 54.76     | 4.98     |
> |HAT| 70.89| 67.12| 10.34| 72.23| 63.89| 3.62| 69.23| 64.23        | 10.67     | 64.98     | 54.23     | 4.10     |
> |HTGNB| 72.12| 67.23| 10.78| 74.23| 64.05| 3.23| 70.56     | 65.32        | 11.12     | 64.87     | 55.20     | 4.45     |
> |HTGNL| 72.78| 68.45| 10.98| 74.89| 64.18| 3.34     | 70.12     | 66.45        | 11.34     | 65.12     | 55.45     | 4.67     |
> |HGWaveNet | 73.12| 68.67| 10.12| 74.12| 64.12| 3.45| 71.05     | 66.78| 11.45     | 65.45| 55.67     | 4.78     |
> |**Our**   | **73.87** | **69.56** | **11.00** | **75.23** | **64.87** | **3.64** | **71.92** | **67.33**    | **11.47** | **66.06** | **56.26** | **5.32** |
>
> - If the CTDG format remains unchanged, the model must incorporate temporal information, such as decay functions or Neural ODEs. This is a key area for improvement.
>
>
> ---
> # C2
> Yes, another common geometric space is the spherical space, which has a constant positive curvature. We replace the hyperbolic space with the spherical space, and the experimental results on two datasets are shown in the table below.
> |||HepPh|||LFB|||HepPh(new)|||LFB(new)||
> |-|-|-|-|-|-|-|-|-|-|-|-|-|
> ||AUC|AP|MRR|AUC|AP|MRR|AUC|AP|MRR|AUC|AP|MRR|
> |Complete Euclidean|96.31|93.86|99.90|91.77|88.93|99.91|96.65|97.11|99.90|92.73|89.12|99.85|
> |Complete Spherical|96.60|93.49|92.85|92.40|88.62|99.85|96.30|96.72|99.85|92.95|88.97|99.89|
> |Our(Hyperbolic+Euclidean)|**97.19**|**94.68**|**99.98**|**93.31**|**89.58**|**99.92**|**97.18**|**97.67**|**99.97**|**93.25**|**89.54**|**99.91**|
>
> The experimental results show that the spherical space does not provide any gain. This may be because spherical space is suitable for graphs with periodic or closed structures, and if the data lacks periodicity or global symmetry, spherical embeddings may not effectively capture the relationships between nodes.
>
> ---
> References:
>
> [1] Foundations and modeling of dynamic networks using dynamic graph neural networks A survey
>
> [2] Towards Better Dynamic Graph Learning: NewArchitecture and Unified Library, NeurIPS 2023
>
> [3] Towards Better Evaluation for Dynamic Link Prediction, NeurIPS 2022

---

> > ### Comment · Reviewer_yHfU · 2025-04-03
> >
> > Thank you for addressing my concerns in detail. The new results are convincing thus, I will raise my score accordingly.

---

> > > ### Author Response · Authors · 2025-04-06
> > >
> > > Thank you for your positive feedback and we are glad to know that our rebuttal and new experiments have addressed most of your concerns!
> > >
> > > The temporal evolution of links in dynamic graphs induces geometric heterogeneity across local structures, making it difficult for a single geometry to model them accurately. Therefore, we propose a cross-geometric framework that unifies Euclidean and hyperbolic spaces through fine-grained substructure modeling. We validate its effectiveness on temporal link prediction and temporal new link prediction tasks.
> > >
> > > We are pleased to learn that you found our paper is well-written and that the experiment design and analysis is sound. In our initial rebuttal, we conducted comparative experiments using the MRR metric, proposed an extension of our method to continuous-time dynamic graphs, and analyzed the experimental results when incorporating another commonly used geometry, namely spherical geometry. We will ensure that these important details are included in the revised version.  Thank you again for your insightful comments and valuable suggestions!
> > >
> > > Best regards,
> > >
> > > Authors

---

### Official Review · Reviewer_Tf9y · 2025-03-15

**Overall Recommendation:** 3

**Summary:**

The paper proposes a unified cross-geometric learning framework that integrates both Euclidean and hyperbolic spaces to address the dynamic nature of graph data. The key idea is to analyze the evolving local structures via k‑hop ego-graphs and select the embedding space that best matches the local geometric characteristics. TSA is introduced to fuse historical states, while a novel LEL is used to finely capture the appearance and disappearance of links over time. Extensive experiments on multiple real-world datasets demonstrate improved performance over state-of-the-art methods in both temporal link prediction and new link prediction tasks.

**Claims And Evidence:**

Yes

**Essential References Not Discussed:**

No

**Experimental Designs Or Analyses:**

Yes

**Methods And Evaluation Criteria:**

Yes

**Other Comments Or Suggestions:**

Refer to weaknesses.

**Other Strengths And Weaknesses:**

Strengths:
1. The paper introduces a novel way to integrate multiple geometric spaces for dynamic link prediction.
2. Extensive experiments and ablation studies provide strong empirical evidence.
3. The framework is comprehensive, covering both spatial and temporal aspects.
Weaknesses:
1. The theoretical contributions are incremental, relying mostly on standard concepts.
2. Some components (like TSA) seem more heuristic than rigorously justified.
3. Additional discussion on hyperparameter sensitivity and potential limitations on scalability would be valuable.
4. Clarity in some figures could be improved(Figures 1 and 3 are cluttered and lack clear legends).

**Questions For Authors:**

1. Can you elaborate on how sensitive the performance is to the choice of k in the k‑hop ego-graph, particularly in graphs with diverse structural properties?
2. How robust is the proposed framework when the underlying graph data is noisy or partially missing? Have you evaluated the model’s performance under such conditions?
3. The TSA layer seems to be a key innovation; could you provide further justification or intuition behind its design and whether it generalizes to other temporal prediction tasks?
4. Could you provide more empirical runtime analysis on significantly larger graphs to validate the computational complexity claims, beyond the datasets used?

**Relation To Broader Scientific Literature:**

The paper is well-situated within the current literature on geometric deep learning and dynamic graph neural networks.

**Theoretical Claims:**

No

---

> ### Author Rebuttal · Authors · 2025-04-01
>
> We sincerely thank the reviewer for the time and effort in reviewing our paper. We take all comments seriously and try our best to address every raised concerns. Please feel free to ask any follow-up questions.
>
> ---
> # No Supplementary Material
> Our appendices include notations, geometric background, experimental details, supplementary experimental results. A zipped code package is available in the Supplementary Material.
>
> ---
> # W4
> Our work has several key innovations. Geometrically, we propose a novel cross-geometric dynamic modeling method by analyzing geometric properties from a local graph perspective in snapshots, advancing the theoretical boundaries of cross-geometric dynamic link prediction. Algorithmically, we design a dynamic link prediction framework with a Temporal State Aggregator layer for efficient historical information extraction. Optimizing from a microscopic viewpoint, we designed a novel Link Evolution Loss function. Practically, our framework achieves state-of-the-art results across various types and scales of continuous graph data.
>
> ---
> # W5 &  Q3
> Since GRU has inherent memory capabilities, the TSA module primarily serves as an auxiliary enhancement. Its design was iteratively refined through extensive experiments. We analyzed and simplified corresponding components in SOTA methods, finding that an attention-based MLP retained the ability to extract key historical information without performance loss while significantly reducing computational complexity, which guided the final design. The table below shows that TSA offers comparable MRR(%) with other modules but with the least time consumption(ms).
> |||HepPh|||LFB||
> |-|-|-|-|-|-|-|
> ||MRR|training|inference|MRR|training|inference|
> |transformer|99.61|60.79|47.26|99.17|19.03|13.62|
> |GRUGCN(counterpart)|98.29|19.66|1.104|98.12|2.273|1.087|
> |HGWaveNet(counterpart)|99.49|1493|11.04|99.32|1021|9.791|
> |Our(TSA)|99.98|17.28|1.047|99.92|1.001|0.999|
>
> ---
> # W6
> We discussed the hyperparameter k (line 430, Figure 6), where k ≥ 4 stabilizes model performance. Figure 7 shows that varying embedding dimensions has minimal impact on the results.
>
> Scalability：
> - Task: Our method can be extended to link prediction on continuous-time dynamic graphs (CTDGs) (Please refer to our responses to **Reviewer yHfU's C1**).
> - Framework: Each component of our model can be extended and upgraded. The results of component replacement are shown below:
> ||HepPh||LFB||
> |-|-|-|-|-|
> | |AUC|AP|AUC|AP|
> |GRU->LSTM|96.82|94.34|92.45|88.96|
> |GCN->GAT|97.47|94.87|93.02|89.15|
> |TSA->Transformer|97.93|95.23|93.86|89.89|
> |Original|97.19|94.68|93.31|89.58|
>
> ---
> # W7
> The legends for Figures 1 and 3 are in the dashed box at the bottom. We apologize for any confusion and will adjust the legend size and provide further explanations if revised.
>
> ---
> # Q1
> In Figure 6 of the paper, we present the AUC and AP values for different tasks on HepPh and LFB datasets as k varies. The results show that when k ≥ 4, the results stabilize, indicating that the geometric features of the local subgraph can be fully described in most cases when k is greater than or equal to 4, which is consistent with the description in work [1].
>
> ---
> # Q2
> We designed experiments to test the model performance under the two scenarios you suggested. The results demonstrate that our method exhibits strong noise resistance and robustness even when some edges are missing. The experimental design is as follows:
> - Noisy(N): Randomly add 5%, 10%, and 20% of non-existing edges to the graph.
> - Partially Missing(M): Randomly remove 5%, 10%, and 20% of the existing edges.
> ||DBLP||Ia-Enron| |DBLP(new)||Ia-Enron(new)||
> |-|-|-|-|-|-|-|-|-|
> ||AUC|AP|AUC|AP|AUC|AP|AUC|AP|
> |Original|93.11|92.32|91.43|89.17|91.23|90.14|89.59|87.16|
> |5%N|90.60|89.69|86.26|79.91|88.63|86.90|85.33|79.17|
> |10%N|88.49|89.50|83.24|76.87|86.82|86.89|81.69|74.83|
> |20%N|82.68|84.86|81.78|77.38|79.65|80.49|79.52|73.77|
> |5%M|92.16|89.79|87.91|81.17|89.57|86.26|88.26|81.84|
> |10%M|91.91|89.15|87.61|79.59|89.37|84.66|88.57|81.06|
> |20%M|92.25|88.17|89.19|80.43|89.36|83.30|89.06|80.80|
>
> ---
> # Q4
> Figure 5 in paper shows the average epoch training and inference time for the Euclidean SOTA method GRUGCN, the hyperbolic SOTA method HGWaveNet, and our method. Following your suggestion, we measured runtime on the largest dataset in [2][3] Contact (2,426,280 dynamic edges) and a million-scale UNVote dataset (1,035,742 dynamic edges). The results (in ms) indicate that our method computational complexity scales linearly while maintaining the least time overhead.
> ||UNVote| |Contact||
> |-|-|-|-|-|
> ||training|inference|training|inference|
> |GRUGCN|726.2|0.538|392.4|3.294|
> |HGWaveNet|7490|16.11|3529|57.41|
> |Our|604.4|0.192|280.9|2.129|
>
> ---
> References:
>
> [1] Tree-like structure in large social and information networks, ICDM 2013
>
> [2] Towards Better Dynamic Graph Learning: NewArchitecture and Unified Library, NeurIPS 2023
>
> [3] Towards Better Evaluation for Dynamic Link Prediction, NeurIPS 2022

---

> > ### Comment · Reviewer_Tf9y · 2025-04-08
> >
> > My concerns have been partially addressed, thereby I have increased the score.

---

> > > ### Author Response · Authors · 2025-04-09
> > >
> > > Thank you for your positive feedback and we are glad to hear that our rebuttal and new experiments have partially addressed your concerns!
> > >
> > > The temporal evolution of links in dynamic graphs induces geometric heterogeneity across local structures, making it difficult for a single geometry to model them accurately. Therefore, we propose a cross-geometric framework that unifies Euclidean and hyperbolic spaces through fine-grained substructure modeling. We validate its effectiveness on temporal link prediction and temporal new link prediction tasks.
> > >
> > > In our initial rebuttal, we primarily provided empirical analyses on the design rationale of the TSA module, evaluated the robustness of our method under noisy or incomplete graphs, and conducted runtime analysis on the larger Contact dataset beyond those used in our paper. We will incorporate these results in the revised version and further explore potential improvements from the perspectives you suggested. Thank you again for your insightful comments and valuable suggestions!
> > >
> > > Best regards,
> > >
> > > Authors

---

### Decision · Program_Chairs · 2025-05-01

**Decision:**

Accept (poster)

**Comment:**

This submission proposes novel methods for analysing dynamic graphs based on _geometrical information_. Essentially, the paper suggests  checking for local hyperbolicity, then mapping features into a hyperbolic space. This is accomplished using the interplay of two modules, viz. a  "Dynamic Geometric Modeling" module, and a "Temporal State Aggregator." The paper demonstrates the utility of this framework on a variety of graph-learning tasks.

After the rebuttal, reviewers were unanimous in their support of the paper, citing in particular the clarity of the writing and comprehensive method description as its strength. I agree with their assessment and am convinced that this submission would make a suitable addition to the conference programme. I trust the authors to implement the changes they promised during the rebuttal and revise their paper accordingly for a camera-ready version.